# Cardiomyocytes stimulate angiogenesis after ischemic injury in a ZEB2-dependent manner

Monika M. Gladka [1], Arwa Kohela[1], Bas Molenaar[1], Danielle Versteeg[1,2], Lieneke Kooijman[1], Jantine Monshouwer-Kloots[1], Veerle Kremer[3,4], Harmjan R. Vos[5], Manon M. H. Huibers [6], Jody J. Haigh[7], Danny Huylebroeck [8,9], Reinier A. Boon[3,10,11], Mauro Giacca[12] & Eva van Rooij [1,2✉]

The disruption in blood supply due to myocardial infarction is a critical determinant for infarct size and subsequent deterioration in function. The identification of factors that enhance cardiac repair by the restoration of the vascular network is, therefore, of great significance. Here, we show that the transcription factor Zinc finger E-box-binding homeobox 2 (ZEB2) is increased in stressed cardiomyocytes and induces a cardioprotective cross-talk between cardiomyocytes and endothelial cells to enhance angiogenesis after ischemia. Single-cell sequencing indicates ZEB2 to be enriched in injured cardiomyocytes. Cardiomyocyte-specific deletion of ZEB2 results in impaired cardiac contractility and infarct healing post-myocardial infarction (post-MI), while cardiomyocyte-specific ZEB2 overexpression improves cardio-myocyte survival and cardiac function. We identified Thymosin β4 (TMSB4) and Prothy-mosin α (PTMA) as main paracrine factors released from cardiomyocytes to stimulate angiogenesis by enhancing endothelial cell migration, and whose regulation is validated in our in vivo models. Therapeutic delivery of ZEB2 to cardiomyocytes in the infarcted heart induces the expression of TMSB4 and PTMA, which enhances angiogenesis and prevents cardiac dysfunction. These findings reveal ZEB2 as a beneficial factor during ischemic injury, which may hold promise for the identification of new therapies.

[1] Hubrecht Institute, Royal Netherlands Academy of Arts and Sciences (KNAW) and University Medical Centre, Utrecht, The Netherlands. [2] Department of Cardiology, University Medical Center, Utrecht, The Netherlands. [3] Department of Physiology, Amsterdam University Medical Center VU, Amsterdam, The Netherlands. [4] Department of Medical Biochemistry, Amsterdam University Medical Center, Amsterdam, The Netherlands. [5] Molecular Cancer Research, Center for Molecular Medicine, University Medical Center, Utrecht, The Netherlands. [6] Department of Pathology, University Medical Centre Utrecht, Utrecht, The Netherlands. [7] Department of Pharmacology and Therapeutics, University of Manitoba, Winnipeg, Canada. [8] Department of Cell Biology, Erasmus University Medical Centre, Rotterdam, The Netherlands. [9] Department of Development and Regeneration, University of Leuven, Leuven, Belgium. [10] Institute for Cardiovascular Regeneration, Centre for Molecular Medicine, Goethe University, Frankfurt am Main, Germany. [11] German Center for Cardiovascular Research (DZHK), Frankfurt am Main, Germany. [12] School of Cardiovascular Medicine and Sciences, King's College London, London, UK. ✉email: e.vanrooij@hubrecht.eu

In the Western world, heart failure has the highest mortality and morbidity rates of all diseases[1]. Ischemic heart disease is caused by the occlusion of a coronary artery, which results in hypoxia-induced loss of cardiomyocytes. The loss of viable myocardium initially induces the activation of fibroblasts required for proper scar formation in the infarcted area. Secondarily to the injury, the remote myocardium undergoes a pathological remodeling response, characterized by cardiomyocyte hypertrophy and further extracellular matrix (ECM) deposition, which can ultimately lead to heart failure and death. Despite the therapeutic benefits of current treatment options for ischemic heart disease, the prevalence continues to increase every year[2]. Therefore, it remains of great importance to gain a better understanding of the mechanisms underlying cardiac remodeling and repair to develop novel and improved therapeutic strategies.

Single-cell sequencing (SCS) has become an attractive tool to study detailed gene expression differences within individual cells in the setting of their local environment. We recently developed a new method to perform SCS in adult cardiac tissue and applied it to healthy and injured hearts. In doing so, we were able to obtain gene expression profiles in specific cell populations coming from the diseased hearts which allowed us to identify novel factors involved in pathological cardiac remodeling[3].

By applying SCS on murine cardiac tissue from both control and injured hearts, we uncovered the DNA-binding transcription factor ZEB2 to be induced in cardiomyocytes that originated predominantly from the injured heart. While ZEB2 has been broadly studied for its role in epithelial to mesenchymal transition[4] and cellular (de)differentiation[5,6], so far, its function in cardiomyocyte biology was unknown. Here we show by gain- and loss-of-function studies that ZEB2 confers cardioprotective effects after ischemia via an increase in circulating levels of TMSB4 and PTMA[7-9]. In line with these findings, the therapeutic delivery of ZEB2 to cardiomyocytes is able to recapitulate these protective effects after MI. These data suggest that the increase of ZEB2 in cardiomyocytes in response to stress protects the heart from ischemic damage via cell non-autonomous effects, which may hold great promise for future heart failure therapies.

## Results

**ZEB2 is expressed in injured cardiomyocytes**. To define novel mechanisms underlying cardiac remodeling and repair after ischemic injury, we subjected mice to either MI or sham surgery ($n = 3$) and collected tissue for single-cell analysis (Fig. 1a–b), applying our method described previously[3]. Using the SORT-Seq protocol[10,11], we obtained the cellular gene expression profiles and detected on average 24,840 transcripts from 2,768 genes per cell. After filtering procedures and quality control measures, we retained a total number of 1186 cells for subsequent analysis (465 from sham and 721 from 3 days post-MI (3 dpMI). K-medoids clustering of 1-Pearson correlation showed that based on gene expression similarities, these cells could be clustered in 14 different cell populations (Fig. 1c and Supplementary Fig. 1a). Using the expression of well-established marker genes, we were able to identify all main cardiac cell types, such as cardiomyocytes, fibroblasts, endothelial cells, and immune cells (Fig. 1c, Supplementary Fig. 1b-g and Supplementary Data 1).

Since cardiomyocytes are critical regulators of cardiac remodeling and repair[12], we decided to focus our analysis on this cell type. Clustering analysis performed only on cardiomyocytes showed that cells originating from sham or injured hearts, for the most part, clustered separately (Fig. 1d), indicating diversity in global gene expression. To explore the biological function of the differentially regulated genes, we subjected enriched genes from injured cardiomyocytes (log2 fold change > 6, $p < 0,001$, $n = 263$ genes) to the PANTHER Classification System[13]. Among others, the enriched genes appeared to encode for cytoskeletal proteins, enzyme modulators, or transcription factors (TFs) (Fig. 1e, Supplementary Data 2). As TFs are potent regulators of gene expression, we set out to explore their functional relevance in more detail. Among the 11 genes known to encode for TFs, we identified Zinc Finger E-box Binding Homeobox2 (ZEB2) (Fig. 1e), which has previously been described to have a role in epithelial to mesenchymal transition and cellular dedifferentiation[14]. Our data indicated ZEB2 to be specifically increased in cardiomyocytes coming from the injured heart (Fig. 1f). Immunofluorescence indicated ZEB2 to be expressed in several cardiac cell types, including macrophages, immune cells, and injured cardiomyocytes flanking the infarcted region (Fig. 1g). Also, in patients suffering from ischemic heart disease, ZEB2 expression could be confirmed in a portion of cardiomyocytes, while we were unable to detect a signal in the control human heart (Fig. 1h).

**Zeb2 deletion from cardiomyocytes impairs cardiac function and repair after MI**. To study the function of ZEB2 in cardiomyocytes, we generated cardiomyocyte-specific Zeb2 knockout mice (*Zeb2* cKO) (Supplementary Fig. 2a). QPCR analysis indicated approximately a 50% reduction in cardiac *Zeb2* expression levels in *Zeb2* cKO mice compared to control (*Zeb2* fl/fl) (Supplementary Fig. 2b). Under baseline conditions deleting *Zeb2* from cardiomyocytes did not appear to induce any gross morphological or functional changes (Supplementary Table 1, and Fig. 2c–g).

Since we found ZEB2 to be upregulated in cardiomyocytes after ischemic stress, we next subjected adult *Zeb2* fl/fl and *Zeb2* cKO mice to sham or MI surgery and collected tissue 14 days later (Fig. 2a). Loss of ZEB2 exclusively from cardiomyocytes was confirmed by qPCR and immunofluorescence (Fig. 2b, c). Compared to the *Zeb2* fl/fl mice, *Zeb2* cKO mice experienced a greater cardiac dysfunction as measured by fractional shortening (FS) and a non-significant increase in left ventricular internal diameter in systole (LVIDs) post-MI (Fig. 2d, e, Supplementary Fig. 3a and Supplementary Table 2). We also observed an increase in infarct size in *Zeb2* cKO compared with *Zeb2* fl/fl mice, assessed by the percentage of the infarcted area of the total LV circumference (Supplementary Fig. 3b-c). These findings were corroborated by histological analysis showing a dilated infarcted region in the *Zeb2* cKO mice compared to controls (Fig. 2f). In line with the observed detrimental consequences of ZEB2 removal from cardiomyocytes, we observed a decrease in the expression of the pro-survival marker BCL-XL after ischemic injury in *Zeb2* cKO mice compared to controls (Fig. 2g–h). Additionally, cardiomyocytes in the remote area displayed a hypertrophic response due to *Zeb2* deletion, as indicated by WGA staining (Fig. 2i–j). Ischemia-induced heart failure is a consequence of impaired perfusion of the heart, and new blood vessel formation is a critical component of wound healing.[15] We observed a significant reduction in the endothelial markers *Pecam1* and *Vegf1* in the infarcted *Zeb2* cKO hearts when compared to *Zeb2* fl/fl controls (Fig. 2k–l), which corresponded with an overall decrease in the percentage of PECAM1 positive vessels in the remote and border zone (Fig. 2m–p). A decrease in capillary density and expression of endothelial markers was also observed in Zeb2 cKO mice after sham, implying a vascular difference at baseline (Supplementary Fig. 3d–j). Together, these data indicate that Zeb2 inhibition in cardiomyocytes impairs cardiac function and promotes pathological remodeling after ischemic injury.

To start exploring the role of ZEB2 in the injured heart in more detail, we performed RNA-seq on cardiac tissue from *Zeb2* fl/fl and *Zeb2* cKO mice subjected to MI. In doing so, we found 2339

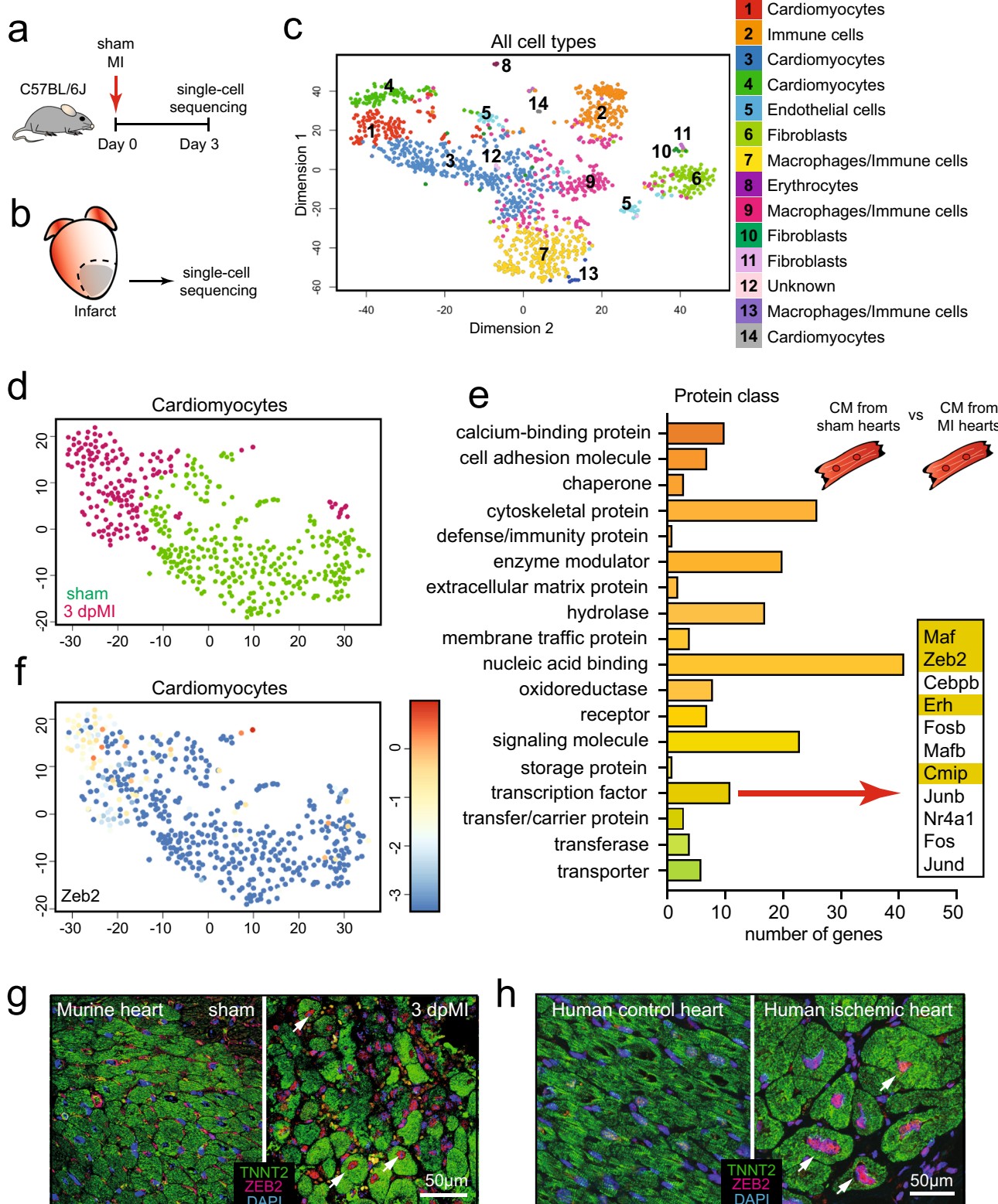

**Fig. 1 ZEB2 is expressed in injured cardiomyocytes. a** Study design. **b** Area of the heart subjected to single-cell sequencing. **c-d** t-SNE map indicating transcriptome similarities among all individual cells. **c** Different numbers and colors highlight different clusters identified by K-medoids clustering. **d** Colors highlight the conditions of the hearts from which the cells were derived (sham in green and 3 dpMI in pink). **e** The PANTHER Classification System analysis on enriched genes from injured cardiomyocytes when compared to sham. **f** t-SNE map showing the expression of *Zeb2* across all cardiomyocytes from sham and 3 dpMI hearts. **g** Representative confocal images from sham and ischemic mouse hearts 3 dpMI stained for ZEB2, ACTN2, and DAPI. **h** Representative confocal images from human control and ischemic hearts stained for ZEB2, ACTN2, and DAPI. Expression is depicted as normalized transcript count on a color-coded scale. t-SNE indicates t-distributed stochastic neighbor embedding. White arrows show ZEB2 positive cardiomyocytes.

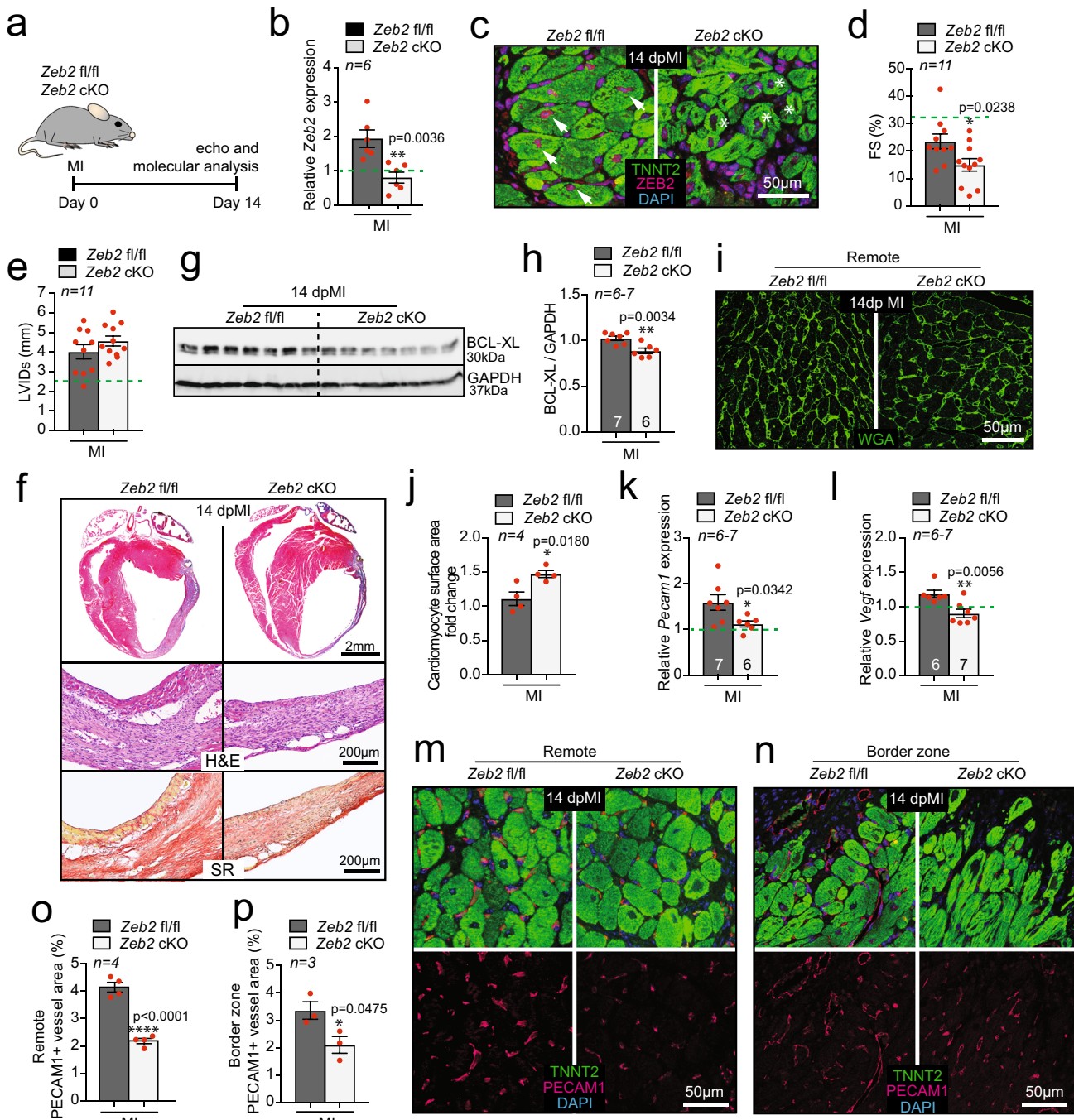

**Fig. 2 Cardiomyocyte-specific deletion of *Zeb2* impairs infarct healing. a** Study design. **b** mRNA expression levels of *Zeb2* in *Zeb2* fl/fl and *Zeb2* cKO mice post-surgery. **c** Immunofluorescence for ZEB2, TNNT2, and DAPI of cardiac sections from *Zeb2* fl/fl and *Zeb2* cKO mice 14 dpMI. **d–e** Quantification of **d** fractional shortening (FS) and **e** left ventricular internal diameter in systole (LVIDs) from *Zeb2* fl/fl and *Zeb2* cKO mice post-surgery. **f** Representative images of cardiac four-chamber view (top panel), sections from infarcted areas stained with either H&E (second panel) or SR (third panel). **g** Representative WB for BCL-XL in tissue from *Zeb2* fl/fl and *Zeb2* cKO 14 dpMI and **h** its quantification. **i** WGA staining to measure cardiomyocyte surface area and **j** its quantification. **k**, **l** mRNA expression levels of **k** *Pecam1* and **l** *Vegf* in *Zeb2* fl/fl and *Zeb2* cKO mice post-surgery. **m**, **n** Immunofluorescence for PECAM1, TNNT2, and DAPI in **m** remote and **n** border zone. **o**, **p** quantification of PECAM1 positive blood vessel areas in histological sections of hearts from *Zeb2* fl/fl and *Zeb2* cKO mice 14 dpMI in (**m**, **n**). Green dashed line indicates sham Zeb2 fl/fl control. H&E indicates Hematoxylin and Eosin, SR indicates Sirius Red. White arrows show ZEB2 positive cardiomyocytes. White stars show ZEB2 negative cardiomyocytes. Data are represented as mean ± SEM, *p < 0.05, **p < 0.01, ****p < 0.0001, each dot indicates a biological replicate, n is indicated in figures. Comparison of two groups was performed with the unpaired, two-tailed Student's *t*-test between Zeb2 cKO vs Zeb2 fl/fl post-MI (**b**, **d**, **e**, **h**, **j**, **k**, **l**, **o**, **p**). Source data are provided as a Source Data file.

(-0,2 Log2 fold change) downregulated and 1599 (0,2 Log2 fold change) upregulated genes (Supplementary Fig. 4a and Supplementary Data 3). The differential expression of the top regulated genes could be confirmed by qPCR in a larger sample set (n = 7)

(Supplementary Fig. 4b-c). Pathway analysis showed that upregulated genes were mainly linked to metabolic pathways (Supplementary Fig. 4d), whereas downregulated genes were involved in, among others, focal adhesion, PI3K-Akt signaling,

and regulation of actin cytoskeleton (Supplementary Fig. 4e), suggesting a decline in cell survival and angiogenic response in *Zeb2* cKO hearts compared to wild type littermates[16–19]. This overrepresentation of pathways involved in ECM and vasculature maintenance indicated that deleting *Zeb2* did not only affect cardiomyocytes but also other cell types, suggesting a cell non-autonomous function mediated by ZEB2 (Supplementary Fig. 4e).

**ZEB2 promotes cross-talk between cardiomyocytes and other cardiac cell types.** Based on the cell non-autonomous effects observed after ZEB2 deletion from cardiomyocytes, we set out to determine whether ZEB2 might be regulating factors that are released from injured cardiomyocytes. To this end, we inhibited ZEB2 in hypoxic neonatal rat cardiomyocytes (NRCMs), after which we collected the conditioned medium and added it on either freshly cultured NRCMs or the non-cardiomyocyte fraction of cells (non-CMs) from the same isolation (Supplementary Fig. 5a). Subsequently, we collected RNA from both conditions and measured the expression of ECM/fibrotic and endothelial markers. We did not observe any significant differences in the cardiomyocyte fraction when exposed to the conditioned medium from treated cardiomyocytes (Supplementary Fig. 5b). However, in using the same conditioned medium, the non-CMs cells did show a decrease in ECM/fibrotic and endothelial genes (Supplementary Fig. 5c).

To study the cellular effects of the conditioned medium in more detail, we exposed both fibroblasts (NIH-3T3) and endothelial cells (HUVECs) to medium from ZEB2-overexpressing cardiomyocytes (H10) (Fig. 3a). To narrow our focus, we fractionated the conditioned medium and screened for the fraction that had the biggest effect on non-CMs populations (Fig. 3b). Treating NIH-3T3 cells and HUVECs with the conditioned medium showed no apparent effects on morphology or activation. Conditioned medium of the ZEB2-overexpressing cardiomyocytes containing the smaller proteins did however induce an increase in endothelial cell migration (Fig. 3c–f) and the expression of angiogenesis-related genes (Fig. 3g–h). Based on these data, we performed mass spectrometry (MS) on the fraction containing proteins of sizes between 3 and 30 kDa. Among the proteins that were detected to be enriched in the supernatant of the ZEB2-overexpressing cells, we identified 40 plasma proteins based on the Human Protein Atlas database https://www.proteinatlas.org/search. Since we were searching for factors directly regulated by ZEB2, out of these 40 plasma proteins, we selected genes that were found to be downregulated in the RNA-seq dataset from the *Zeb2* cKO hearts ($n = 11$) (Fig. 3i and Supplementary Data 4). Overexpression or knockdown of *Zeb2* (Fig. 3j–k), resulted in an increase or decrease of the 11 identified factors, respectively (Fig. 3l–m), indicating a regulatory effect of ZEB2 on the expression of these genes. Together these data suggest that ZEB2 regulates the secretion of factors from cardiomyocytes that directly affect endothelial cell migration.

**Pro-angiogenic factors TMSB4 and PTMA are regulated by ZEB2.** To start exploring the direct effect of ZEB2 on these circulating factors in vivo, we first set out to measure their expression levels in *Zeb2* cKO mice subjected to MI. While the expression of many factors did not change in *Zeb2* cKO mice post-MI, we observed a significant downregulation for *Tmsb4, Ptma, Rack1* and *Dynll1* (Fig. 4a–b, Supplementary Fig. 6a–i). Interestingly, *Tmsb4* and *Ptma* are actively secreted factors that have previously been associated with endothelial cell migration and angiogenesis[7,8,20]. Immunofluorescence validated a reduction of TMSB4 and PTMA protein in injured *Zeb2* cKO hearts 14 days post-MI (14 dpMI) (Fig. 4c–d). As we saw an induction of ZEB2

after ischemic injury in mice, we next measured whether this was also true for *Tmsb4* and *Ptma*. All three genes showed a transcriptional activation post-MI with a peak in expression 3-7 days after injury (Fig. 4e–g). This positive correlation between the expression of ZEB2 and the *TMSB2* and *PTMA* could also be confirmed in cardiac samples from patients suffering from ischemic heart disease (Fig. 4h–i), suggesting a conserved mechanism. These data imply a possible regulation of *Tmsb4* and *Ptma* by ZEB2 most likely due to multiple ZEB2 binding sites present in close proximity to the transcription starting site (TSS) in *Tmsb4* and *Ptma* in mouse and human (Supplementary Fig. 6j). Additionally, we made use of existing ChIP-seq data (https://www.encodeproject.org/targets/ZEB2-human/) and identified ZEB2 binding motifs in enhancer regions of both TMSB4 and PTMA (Supplementary Fig. 6k-l), indicating that ZEB2 protein can directly bind to those enhancers causing an increase in TMSB4 and PTMA expression.

**TMSB4 and PTMA from cardiomyocytes drive endothelial cell migration and proliferation.** To examine the secretory and pro-angiogenic effect of TMSB4 and PTMA from cardiomyocytes, we infected human iPSC-derived cardiomyocytes with adeno-associated viruses overexpressing either ZEB2, TMSB4 or PTMA in the absence or presence of a siRNA targeting ZEB2 or a control (Supplementary Fig. 7a). This resulted in an increase or decrease in expression compared to control cells, as indicated by qPCR analysis (Supplementary Fig. 7b-e). Conditioned medium from cardiomyocytes overexpressing ZEB2, TMSB4 or PTMA enhanced the migration of endothelial cells, with no apparent additive effect when overexpressing both TMSB4 or PTMA (Supplementary Fig. 7f-g). Conversely, exposing endothelial cells to conditioned medium from iPSC-derived cardiomyocytes treated with a siRNA against ZEB2, lowered endothelial cell migration (Supplementary Fig. 7h-i). However, this inhibitory effect could be rescued by the conditioned medium from cardiomyocytes, where ZEB2 inhibition was followed by over-expression of TMSB4 or PTMA (Supplementary Fig. 7h-i). Additionally, we performed sprout formation and proliferation assays to further validate the importance of ZEB2 in different aspects of the angiogenic response. To this end, we treated iPSC-derived cardiomyocytes with siRNA-control, siRNA-ZEB2, siRNA-TMSB4, siRNA-PTMA and a combination of siRNA-TMSB4 and siRNA-PTMA (Fig. 5a). Next, we collected conditioned medium from treated cardiomyocytes and used it to treat HUVECs (Fig. 5a). In doing so observed that ZEB2 inhibition in cardiomyocytes resulted in a significant decrease in the number of sprouts per spheroid and a non-significant decrease in cumulative sprout length per spheroid (Fig. 5b–d). Inhibition of TMSB4 and PTMA separately did not affect the number and length of sprouts, but when inhibited together, it showed a comparable effect as ZEB2 inhibition (Fig. 5b–d). We also analyzed endothelial cell proliferation and observed a significant decrease in EdU positive nuclei when treated with conditioned medium from ZEB2-inhibited cardiomyocytes (Fig. 5e–f). Even though we did not reach significance in the decrease of EdU positive HUVECs in TMSB4 and PTMA-treated conditions, we did observe a very pronounced decrease in the expression of cell cycle genes in all conditions (Fig. 5g–i). Conversely, when we used conditioned medium from cardiomyocytes overexpressing ZEB2, TMSB4, PTMA and a combination of TMSB4 and PTMA (Fig. 6a), we observed that overexpression of ZEB2 in cardiomyocytes increased sprout number and cumulative sprout length per spheroid in the presence of VEGF (Fig. 6b–d). Conditioned medium from TMSB4-expressing cardiomyocytes enhanced those effects even without VEGF treatment (Fig. 6b–d). Additionally, all

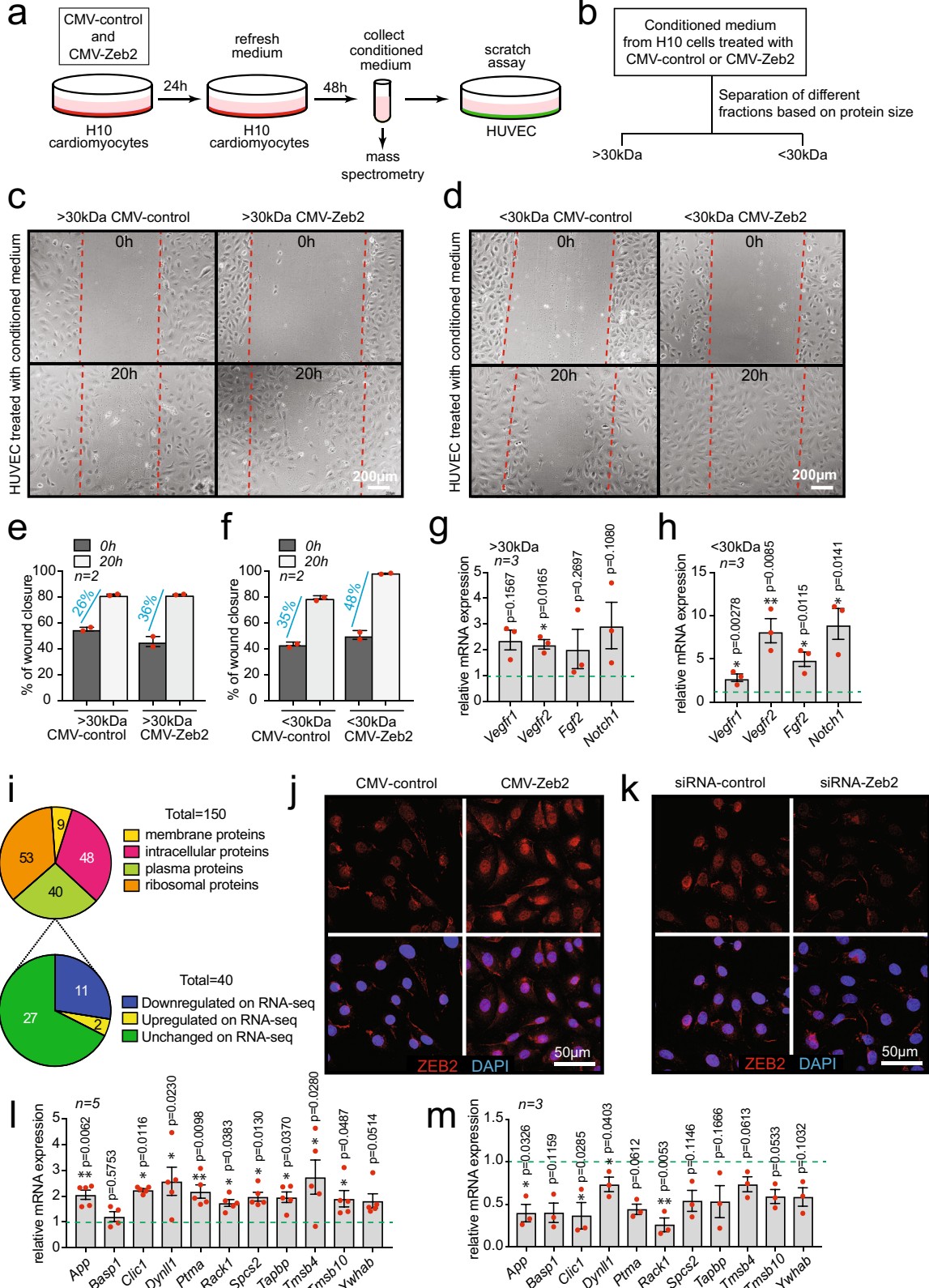

treatments were able to increase proliferation in HUVECs, as indicated by EdU incorporation (Fig. 6e–f).

These findings support the notion that ZEB2 mediates endothelial cell migration and proliferation, at least partially via the secretion of TMSB4 or PTMA from cardiomyocytes.

**Genetic overexpression of ZEB2 protects the heart from ischemic damage and promotes an angiogenic response.** Based on the detrimental phenotype observed after ZEB2 removal from cardiomyocytes, we generated cardiomyocyte-specific ZEB2 overexpressing mice (*Zeb2* cTg). To do so, we crossed Rosa26-

**Fig. 3 Mass spectrometry to identify ZEB2 targets. a** Experimental setup. **b** Fractionation strategy of conditioned medium for mass spectrometry purposes. **c-d** Representative images of scratch assay in HUVECs treated with the indicated conditioned media at 0 and 20 hours after treatment. **e-f** Quantification of wound closure from (**c-d**). **g-h** mRNA expression levels of *Vegfr1, Vegfr2, Fgf2,* and *Notch1* in HUVECs after treatment with conditioned medium. **i** Pie charts showing numbers of proteins identified by mass spectrometry in fraction 3-30 kDa. **j** ZEB2 overexpression in H10 cells and **k** ZEB2 inhibition in NRCMs. **l-m** Validation of mRNA expression of genes encoding for the 11 selected proteins identified by mass spectrometry in (**l**). Green dashed line indicates corresponding control. Data are represented as mean ± SEM, *p < 0.05, **p < 0.01, each dot indicates a biological replicate, n is indicated in figures. Comparison of two groups was performed with the unpaired, two-tailed Student's *t*-test compared to the control indicated as green dashed line (**g**, **h**, **l**, **m**). Source data are provided as a Source Data file.

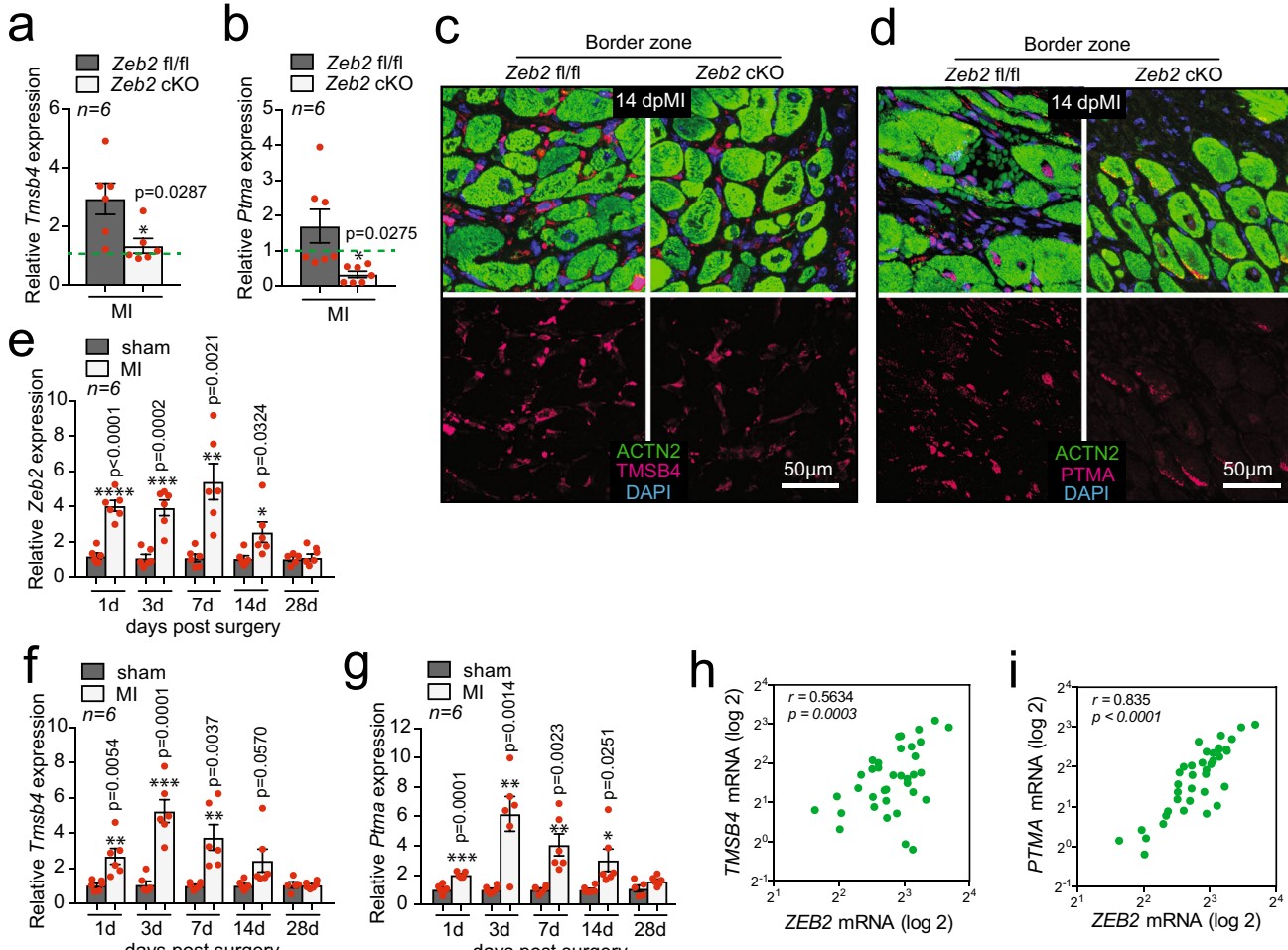

**Fig. 4 *Tmsb4* and *Ptma* as ZEB2 target genes. a-b** mRNA expression levels of **a** *Tmsb4* and **b** *Ptma* in *Zeb2* fl/fl and *Zeb2* cKO mice post-surgery. **c-d** Immunofluorescence for **c** TMSB4, ACTN2, and DAPI and **d** PTMA, ACTN2, and DAPI of histological sections of hearts from *Zeb2* fl/fl and *Zeb2* cKO mice 14 dpMI. **e-g** mRNA expression levels of **e** *Zeb2*, **f** *Tmsb4,* and **g** *Ptma* in mice subjected to MI for different time periods. **h-i** 1-Pearson correlation between *ZEB2* and **h** *TMSB4* and **i** *PTMA* in human ischemic hearts. Green dashed line indicates sham Zeb2 fl/fl control. Data are represented as mean ± SEM, *p < 0.05, **p < 0.01, ***p < 0.001, ****p < 0.0001, each dot indicates a biological replicate, n is indicated in figures. Comparison of two groups was performed with the unpaired, two-tailed Student's *t*-test, between Zeb2 cKO vs Zeb2 fl/fl post-MI (**a**, **b**) and between sham vs MI at different time points (**e**, **f**, **g**). Source data are provided as a Source Data file.

LoxSTOPLox-ZEB2 mice[21] with mice overexpressing Cre recombinase under the control of the cardiac-specific α-myosin heavy chain promoter (αMHC-Cre Tg)[22] (Supplementary Fig. 8a-b). Overexpression of *Zeb2* was confirmed by qPCR for *Zeb2* (and *eGFP*) (Supplementary Fig. 8c-d). Compared to WT littermates, we did not observe any morphological or functional effects of ZEB2 overexpression under baseline conditions (Supplementary Fig. 8e–g and Supplementary Table 3).

To study the function of ZEB2 overexpression during an ischemic event, we subjected *Zeb2* cTg mice to MI for 14 days, after which we performed functional and molecular analyses

(Fig. 7a). *Zeb2* cTg-based overexpression was confirmed on mRNA (Fig. 7b–c) and protein level (Fig. 7d, Supplementary Fig. 9a). Echocardiography showed an improvement in cardiac function (FS), a decline in left ventricular internal diameter (LVIDs), and smaller infarct size in mice overexpressing ZEB2 in cardiomyocytes (Fig. 7e–f, Supplementary Fig. 9b–d and Supplementary Table 4). This was confirmed by histology showing better maintenance of cardiac morphology of the infarcted region in the *Zeb2* cTg mice with a reduction in the fibrotic response (Fig. 7g).

Contrary to the observations made in the *Zeb2* cKO mice, we detected a significant increase in *Tmsb4 and Ptma* in response to

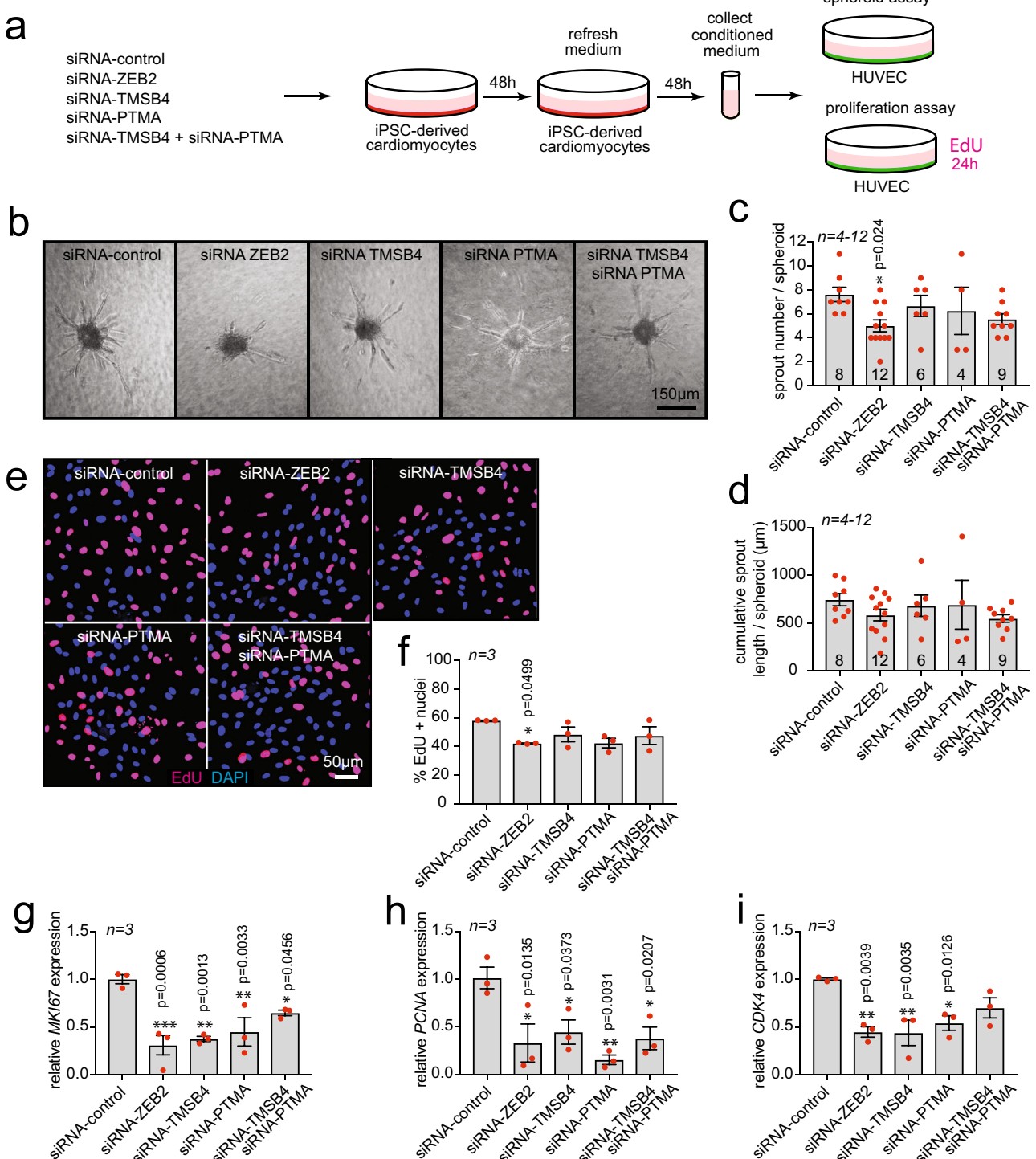

**Fig. 5 Knock-down of ZEB2, TMSB4, and PTMA in cardiomyocytes inhibits endothelial cell migration and proliferation. a** Experimental setup. **b** Representative images of sprouts formed by HUVECs upon treatment indicated in (**a**). **c** Quantification of sprouts number per spheroid. **d** Quantification of cumulative sprout length per spheroid. Data in **b**-**d** is from one experiment with 4-12 biological replicates. **e** Representative immunofluorescent images of HUVECs treated as indicated in (**a**), nuclei stained for DAPI and EdU. **f** Quantification of EdU positive nuclei. **g-i** mRNA expression levels of **g** Mki67, **h** Pcna **i** Cdk4 in HUVECs treated with conditioned medium from (**a**). Data are represented as mean ± SEM, each dot indicates a biological replicate, n is indicated in figures. Data were analyzed by ordinary one-way ANOVA with Dunnett's multiple comparison test (**c**, **d**, **f**, **g**, **h**, **i**). Source data are provided as a Source Data file.

ZEB2 overexpression (Fig. 7h–k). This corresponded with an increase in *Pecam1* at both the mRNA and protein levels (Fig. 7l–n) and an overall increase in the PECAM1 positive area in the *Zeb2* cTg hearts in the remote and border zone (Fig. 7o–p). Additionally, we already observed a mild increase in capillary density and endothelial markers under sham conditions, which can potentially contribute to the enhanced angiogenesis post-MI (Supplementary Fig. 9e-k). In line with the protective consequences of overexpressing ZEB2 in cardiomyocytes, we observed an increase in the expression of the pro-survival gene

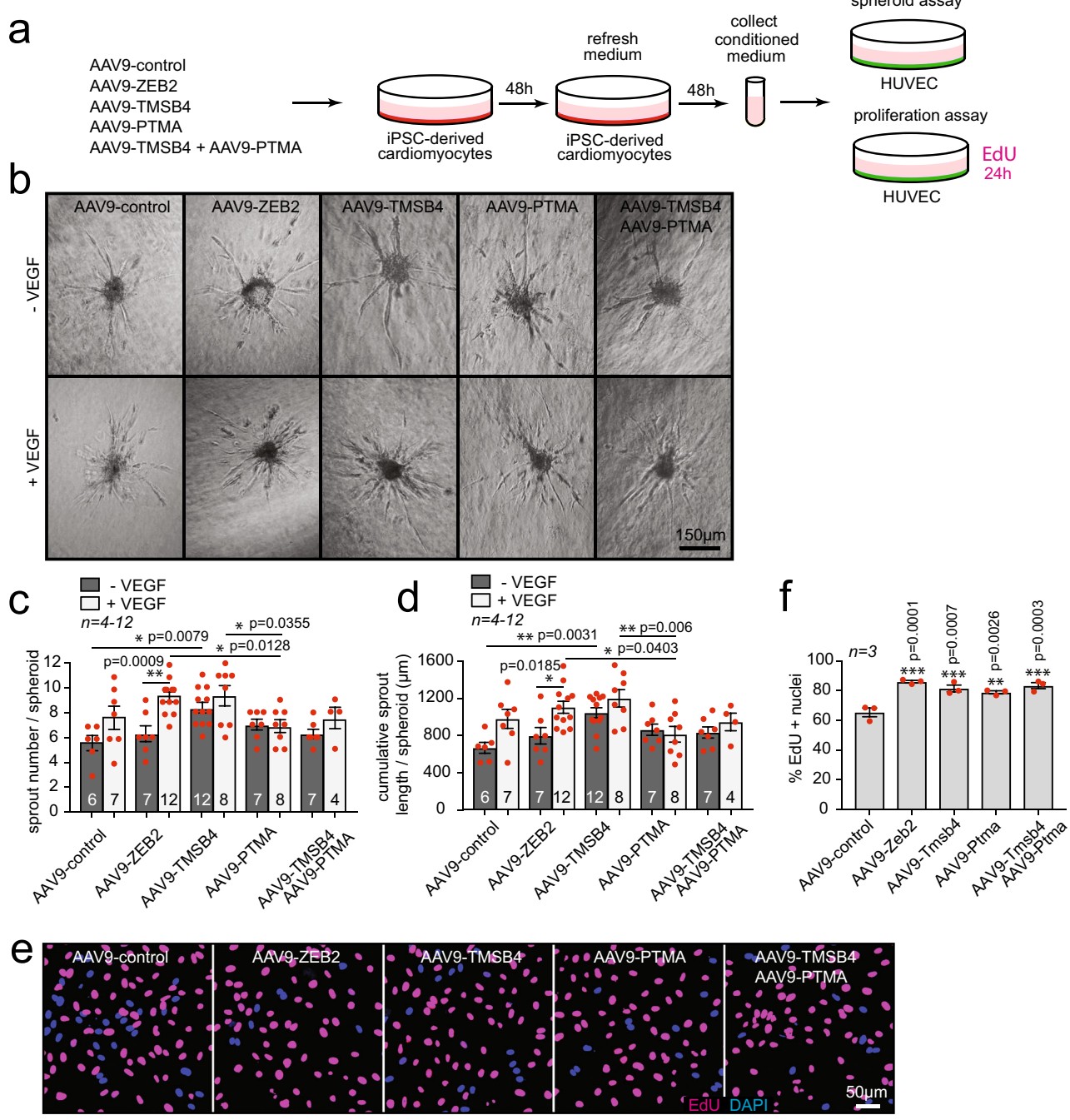

**Fig. 6 Overexpression of ZEB2, TMSB4, and PTMA in cardiomyocytes enhances endothelial cell migration and proliferation. a** Experimental setup. **b** Representative images of sprouts formed by HUVECs upon treatment indicated in (**a**). **c** Quantification of sprouts number per spheroid. **d** Quantification of cumulative sprout length per spheroid. Data in **b**-**d** is from one experiment with 4-12 biological replicates. **e** Representative immunofluorescent images of HUVECs treated as indicated in (**a**), nuclei stained for DAPI and EdU. **f** Quantification of EdU positive nuclei. Data are represented as mean ± SEM, *$p <$ 0.05, **$p <$ 0.01, ***$p <$ 0.001, each dot indicates a biological replicate, n is indicated in figures. Data were analyzed by ordinary one-way ANOVA with Dunnett's multiple comparison test (**f**) and with two-way ANOVA with Sidak's multiple comparison test (**c**, **d**). Source data are provided as a Source Data file.

BCL-XL in *Zeb2* cTg mice post-MI (Supplementary Fig. 9l-m). This corresponded to a reduction in cardiomyocyte hypertrophy in the remote area of the *Zeb2* cTg hearts, as indicated by WGA staining (Fig. 7q–r).

Together these data demonstrated that transgenic overexpression of ZEB2 in cardiomyocytes induces a cardioprotective response due to an improved angiogenesis post-MI.

**AAV9-mediated ZEB2 delivery is therapeutically beneficial**. To explore the increase in ZEB2 in a therapeutically relevant manner, we made use of AAV9-mediated gene transfer, previously reported to be a useful tool for local delivery to cardiomyocytes.[23] We generated AAV9-control and AAV9-Zeb2 (Supplementary Fig. 10a) and tested different routes of delivery, including intra-peritoneal (IP) injection into mouse pups, and intracardiac (IC)

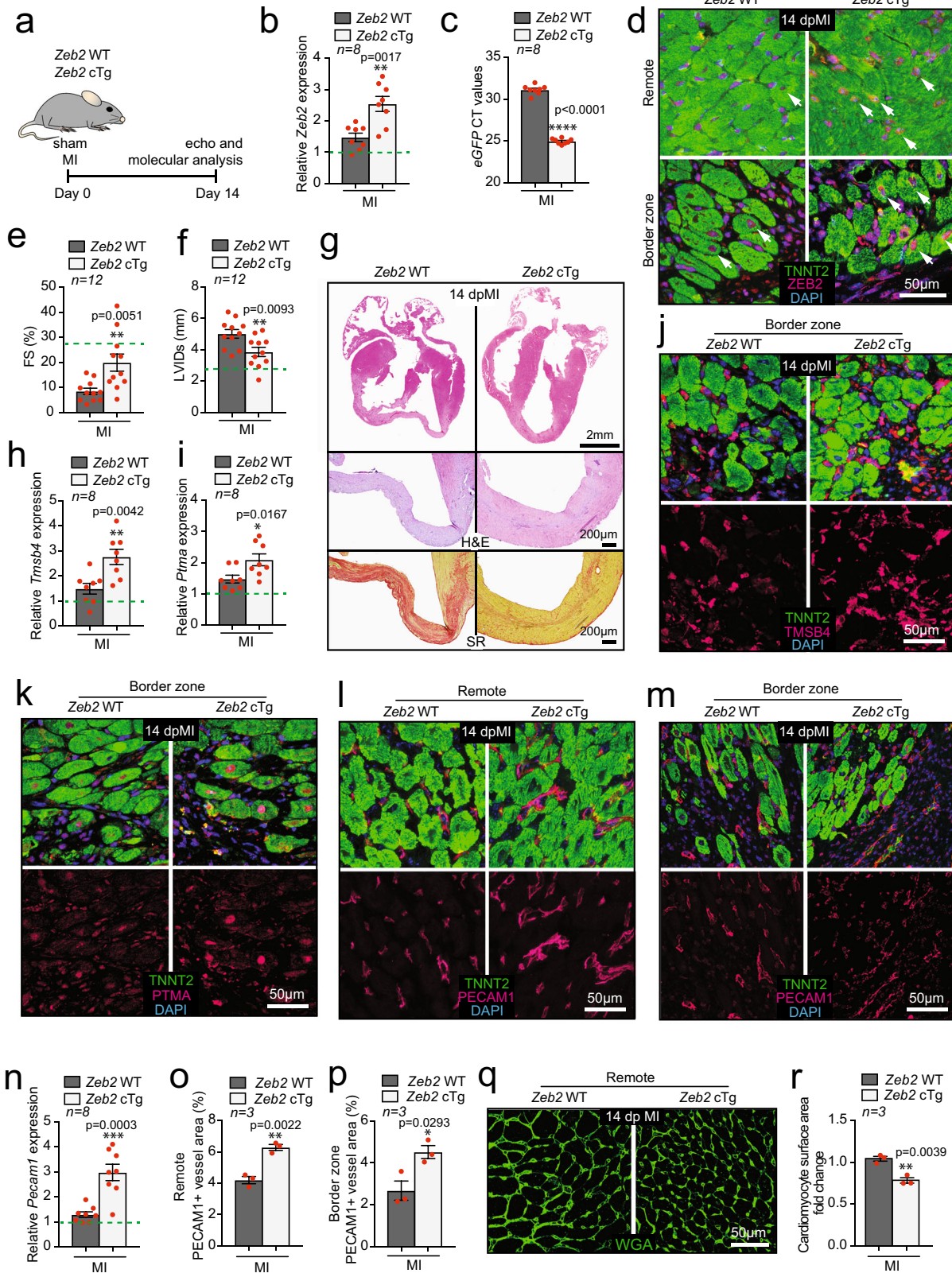

and intravenous (IV) injection into adult mice (Supplementary Fig. 10b–d). Since intracardiac injection resulted in the strongest increase in expression of *Zeb2* in the heart (Supplementary Fig. 10c), we used this method for our subsequent experiments.

To test the therapeutic benefit of intracardiac delivery of ZEB2, we injected male C57BL/6 J mice immediately after sham or MI

surgery with AAV9-control or AAV9-Zeb2 and performed the analysis 14 and 28 days later (Fig. 8a). ZEB2 expression was confirmed to be increased at both times points post-injection (Fig. 8b–d and Supplementary Fig. 11a–c). Increasing ZEB2 in cardiomyocytes improved cardiac function and morphology after MI at both time points as measured by FS and LVID (Fig. 8e–f

**Fig. 7 Genetic overexpression of ZEB2 protects the heart from ischemic damage. a** Study design. **b-c** mRNA expression level of **b** *Zeb2*, **c** *eGFP* (CT values) in *Zeb2* WT and *Zeb2* cTg mice post-surgery. **d** Immunofluorescence for ZEB2, TNNT2 and DAPI of histological sections of hearts from *Zeb2* WT and *Zeb2* cTg mice 14 dpMI. **e-f** Quantification of **e** fractional shortening (FS) and **f** left ventricular internal diameter in systole (LVIDs) from *Zeb2* WT and *Zeb2* cTg mice post-surgery. **g** Representative images of cardiac four-chamber view (top panel) and sections of infarcted areas stained with H&E (second panel) or SR (third panel). **h, i** mRNA and **j, k** protein expression levels of *Tmsb4 and Ptma* in *Zeb2* WT and *Zeb2* cTg mice post-surgery. **l-m** Immunofluorescence for PECAM1, TNNT2, and DAPI in l remote and m border zone in hearts from *Zeb2* WT and *Zeb2* cTg mice post-MI. **n** mRNA expression levels of *Pecam1* in hearts from *Zeb2* WT and *Zeb2* cTg mice post-surgery. **o-p** quantification of PECAM1 positive blood vessel area in histological sections from (**l-m**). **q** WGA staining to measure cardiomyocyte surface area and **r** its quantification. Green dashed line indicates sham Zeb2 WT control. H&E indicates Hematoxylin and Eosin, SR indicates Sirius Red. White arrows show ZEB2 positive cardiomyocytes. Data are represented as mean ± SEM, *$p < 0.05$, **$p < 0.01$, ***$p < 0.001$, each dot indicates a biological replicate, n is indicated in figures. Comparison of two groups was performed with the unpaired, two-tailed Student's *t*-test between Zeb2 cTg vs Zeb2 WT post-MI (**b, c, e, f, h, i, n, o, p, r**). Source data are provided as a Source Data file.

and Supplementary Fig. 11d-e, Supplementary Table 5, Supplementary Table 6). We detected a significant increase in *Tmsb4 and Ptma* levels in response to ZEB2 delivery (Fig. 8g–j and Supplementary Fig. 11f–i). This corresponded with an increase in *Pecam1* and *Vegf* expression (Fig. 8k–m and Supplementary Fig. 11j–l) and an overall increase in the PECAM1 positive area in *Zeb2* cTg hearts (Fig. 8n and Supplementary Fig. 11m). Additionally, when increasing ZEB2, we observed a trend in elevated levels of the pro-survival gene BCL-XL 28 days post- MI (Fig. 8o–p) and smaller cardiomyocyte size in the remote area as indicated by WGA staining (Fig. 8q–r and Supplementary Fig. 11n-o). Our data suggest that the therapeutic delivery of ZEB2 stimulates cardiomyocyte-derived angiogenic signals in the injured heart and thereby contributes to improved cardiac repair and cardiac function (Fig. 9).

## Discussion

Heart disease continues to be the leading cause of death worldwide, due to limited therapeutic options and the heart's inability to regenerate healthy cardiomyocytes after myocardial infarction. Discovery of players involved in myocardial repair after ischemia could serve in the development of enhanced therapeutic interventions.

Here we use single-cell sequencing to show that ZEB2 is expressed in injured cardiomyocytes, where it functions as a cardioprotective factor involved in infarct healing and cardiac repair in a non-autonomous manner. ZEB2 expression was increased in the heart only during the first days after MI, the time when endogenous repair response takes place, and before a fibrotic scar is being produced[24]. Constitutive expression of ZEB2, as well as its therapeutic delivery to the injured heart, triggered a subsequent increase in vessel density resulting in diminished scar formation and preserved cardiac function. Our data demonstrate the importance of cardiomyocyte signaling after ischemic injury and, for the first time, show the importance of ZEB2 during cardiac repair.

While ZEB2 as a transcription factor is well known for its functions in EMT and development[4,25], it has only been shown to function in a cell non-autonomous manner in brain cortex development[26]. This study demonstrated that ZEB2 regulates the production of signaling factors in post-mitotic neurons, which feedback to progenitors to influence the timing of cell fate switch and the number of neurons and glial cells throughout corticogenesis[26].

Cardiac cell populations interact with each other not only by physical contact, receptor-ligand interactions, or cell adhesion complexes, but also via a variety of soluble paracrine, autocrine, and endocrine factors[27]. Secreted proteins are required to maintain normal cardiac function and control pathological remodeling of the myocardium in response to injury. Our data show that ZEB2 expression in cardiomyocytes induces the expression and

secretion of factors involved in cardiac remodeling post-MI. Mass spectrometry on conditioned medium from cardiomyocyte-like cells overexpressing ZEB2 revealed an increase in circulating levels of TMSB4 and PTMA, which have already been linked to cardioprotection by regulating neovascularisation[8], angiogenesis[28], and apoptosis[29], and could explain the observed in vivo phenotype. TMSB4 treatment is already being used to promote wound repair in skin, cornea, and heart in a series of ongoing clinical trials[30]. In the heart, studies have shown that TMSB4 stimulates the formation of new cardiomyocytes, which originate from the epicardial layer[31], and promotes neovascularization[32]. Additionally, TMSB4 has been shown to play anti-inflammatory, antioxidant, and antifibrotic effects during liver injury in mice[33]. On the other hand, PTMA has been shown to stimulate cardiac endothelial cell migration, angiogenesis, and wound healing[20]. It has also been recently proven to protect from retinal ischemic damage[34]. Here we showed that both TMSB4 and PTMA are transcriptionally regulated by ZEB2 in injured cardiomyocytes and are subsequently secreted to promote downstream protective signaling. Recent reports on ZEB2 also suggested that it promotes cell survival and angiogenesis in cancer[35], which is in line with the biological functions we observed in the injured heart.

Our results present ZEB2 as a promising mediator of angiogenic signals coming from cardiomyocytes in response to injury. Although genetic deletion and overexpression of ZEB2 in cardiomyocytes have minor effects on capillary density under baseline conditions, therapeutic delivery of ZEB2 post-injury is sufficient to restore cardiac function by stimulating angiogenesis, which in turn improves cardiac repair. Targeted delivery using a cardiomyocyte-specific promoter would further improve the specificity of the therapy and allow for a more directed treatment in order to prevent undesired effects in other organs. Although follow-up studies will have to provide more detailed insights into the exact mechanisms, both the identification of ZEB2 and its downstream signals offer potential new therapeutic opportunities for patients suffering from ischemic heart disease.

## Methods

**Mice.** All mouse studies were conducted in accordance with protocols approved by the ethics committee of the Hubrecht Institute in Utrecht. Mice were housed in a normal condition with 12:12 h light: dark cycle in a temperature-controlled room with food and water ad libitum. For all animal experiments, we used 8–9 weeks old male mice.

*Myocardial infarction (MI) and transthoracic echocardiography (echo).* Myocardial infarction or sham surgery was performed in C57BL/6 J wild type mice (Figs. 1 and 7), *Zeb2* fl/fl and *Zeb2* cKO mice (Fig. 2), *Zeb2* WT and *Zeb2* cTg (Fig. 6) (8-10 weeks of age) by permanent ligation of the left anterior descending artery (LAD) for 3 days (Fig. 1 and corresponding Supplementary figures), 14 days (Figs. 2, 6 and corresponding Supplementary figures) or 28 days (Fig. 7 and corresponding Supplementary figures). In brief, mice were anesthetized with a mixture of ketamine and xylazine by IP injection, and hair was shaved from the thorax. A tracheal tube

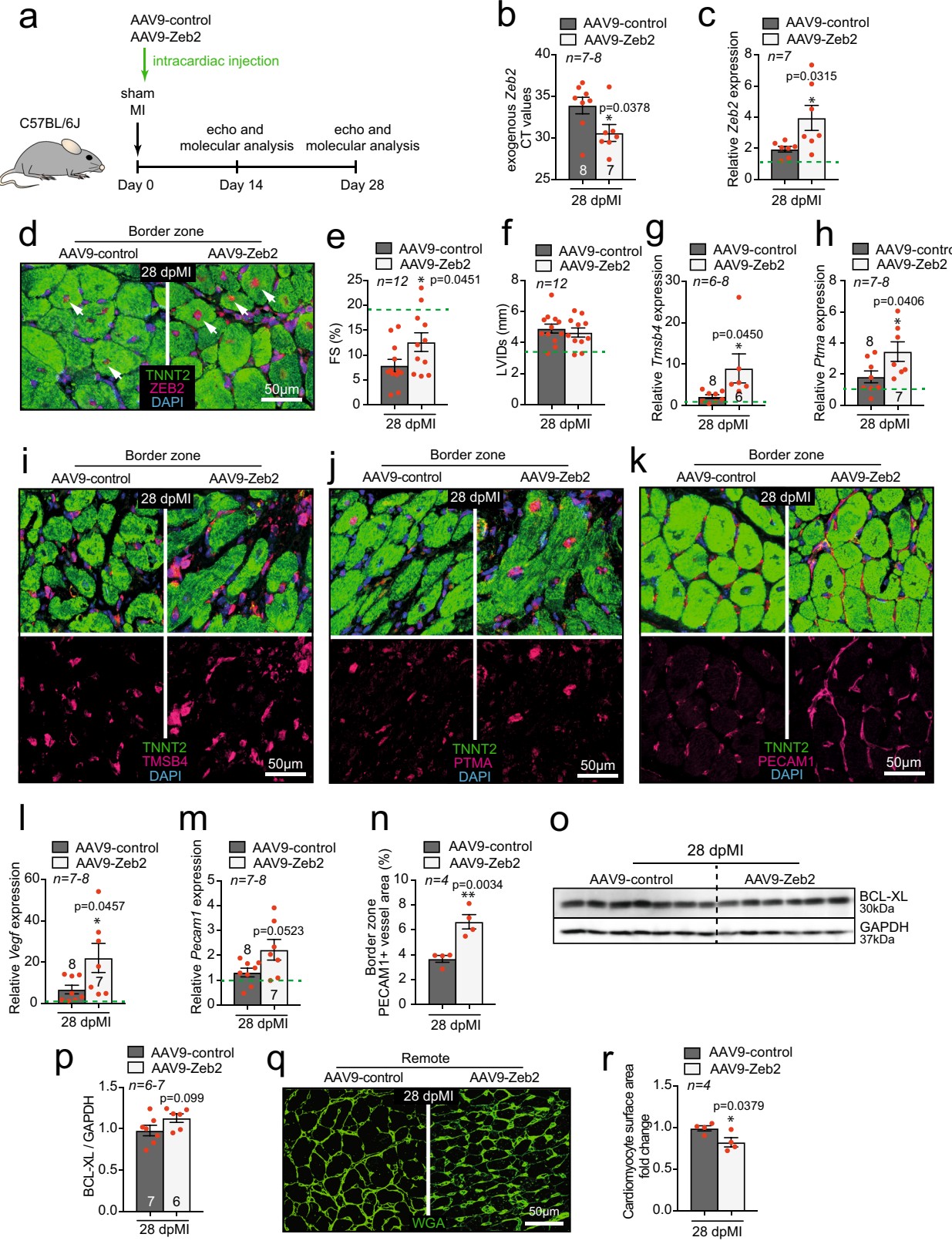

was placed and the mouse was connected to a ventilator (UNO Microventilator UMV-03, Uno BV.). The surgical site was cleaned with iodine and 70% ethanol. Using aseptic technique with sterile instruments the skin was incised of the midline to allow access to the left third intercostal space. Pectoral muscles were retracted and the intercostal muscles were cut caudal to the third rib. Wound hooks were placed to allow access to the heart. The pericardium was incised longitudinally and the left anterior descending coronary artery (LAD) identified. A 7.0 silk suture was placed around the LAD. Successful ligation of the coronary artery was definite by whitening of the heart anterior wall. After the surgery, the rib cage was closed with 5.0 silk suture, and the skin closed with a wound clip. The animal was disconnected from the ventilator, the tracheal tube removed, and placed on a nose cone with 100% oxygen. To alleviate pain or discomfort after the surgery, mice were injected with 0.05–0.1 mg/kg of Buprenorphine. During the whole procedure and recovery period animals were placed on a 38 °C hot plate. Cardiac function was evaluated by

**Fig. 8 AAV9-mediated ZEB2 cardiac delivery improves infarct healing and cardiac function 28 dpMI. a** Study design. **b-c** Expression levels of **b** exogenous *Zeb2* (CT values) and **c** *Zeb2* mRNA in AAV9-control and AAV9-Zeb2 treated mice post-surgery. **d** Immunofluorescence for ZEB2, TNNT2 and DAPI of histological sections of hearts from AAV9-control and AAV9-Zeb2 treated mice 28 dpMI. **e-f** Quantification of **e** fractional shortening (FS) and **f** left ventricular internal diameter in systole (LVIDs) from AAV9-control and AAV9-Zeb2 treated mice post-surgery. **g-h** mRNA expression levels of **g** *Tmsb4* and **h** *Ptma* in hearts from AAV9-control and AAV9-Zeb2 treated mice 28 dpMI. **i-k** Immunofluorescence for **i** TMSB4, **j** PTMA and **k** PECAM1, TNNT2 and DAPI of histological sections of hearts from AAV9-control and AAV9-Zeb2 treated mice 28 dpMI. **l-m** mRNA expression levels of **l** *Vegf* and **m** *Pecam1* in hearts from AAV9-control and AAV9-Zeb2 treated mice 28 dpMI. **n** Quantification of PECAM1 positive blood vessel areas in histological sections of hearts from AAV9-control and AAV9-Zeb2 treated mice 28 dpMI. **o** Representative WB for BCL-XL in tissue from AAV9-control and AAV9-Zeb2 treated mice 28 dpMI and **p** its quantification. **q** WGA staining to measure cardiomyocyte surface area and **r** its quantification. Green dashed line indicates sham AAV9-control. White arrows show ZEB2 positive cardiomyocytes. Data are represented as mean ± SEM, *$p < 0.05$, **$p < 0.01$, ***$p < 0.001$, ****$p < 0.0001$, each dot indicates a biological replicate, n is indicated in figures. Comparison of two groups was performed with the unpaired, two-tailed Student's *t*-test between AAV9-Zeb2 vs AAV9-control post-MI (**b**, **c**, **e**, **f**, **g**, **h**, **l**, **m**, **n**, **p**, **r**). Source data are provided as a Source Data file.

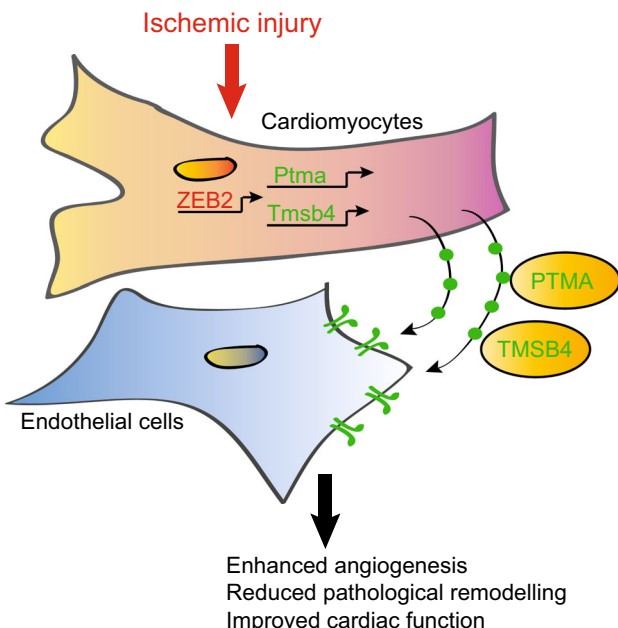

**Fig. 9 ZEB2 expression in injured cardiomyocytes promotes angiogenic signaling.** Model depicting the activation of ZEB2 in injured cardiomyocytes, resulting in transcriptional activation of *Tmsb4* and *Ptma* followed by their secretion from cardiomyocytes subsequently enhancing the angiogenic response.

two-dimensional transthoracic echocardiography on sedated, adult mice (2% iso-flurane) using a Visual Sonic Ultrasound system with a 30 MHz transducer (VisualSonics Inc., Toronto, Canada). The heart was imaged in a parasternal long-axis as well as short-axis view at the level of the papillary muscles, to record M-mode measurements, determine heart rate, wall thickness, and end-diastolic and end-systolic dimensions. Fractional shortening (defined as the end-diastolic dimension minus the end-systolic dimension normalized for the end-diastolic dimension) as well as ejection fraction (defined as the stroke volume normalized for the end-diastolic volume), were used as an index of cardiac contractile function. Of note, MI surgeries were performed by two different animal surgeons and the severity of the MI differs between studies, however the comparison is always performed within the studies.

*Digestion of the heart and single-cell sorting.* Single-cell experiments were performed as described previously[1]. In short, mice (3 days post sham or post-MI) were euthanized and hearts were perfused with cold perfusion buffer (135 mM NaCl, 4 mM KCl, 1 mM MgCl2, 10 mM HEPES, 0.33 mM NaH2PO4,10 mM glucose, 10 mM 2,3-butanedione monoxime (Sigma, St Louis, MO), 5 mM taurine (Sigma), adjust pH to 7.2 at 37 °C. The infarcted area of the heart (infarct and border zone) or the corresponding region of sham hearts (*n* = 3 per condition) were collected and minced into small pieces using a scalpel and transferred into a glass vial with 1.5 ml of cold digestion buffer (0.5 mg/ml Liberase TL (Roche, #5401020001), 20μg/ml DNase I (Worthington, LK003172), 1 M HEPES, Dulbecco's Modified Eagle Medium (DMEM), high glucose, GlutaMAX™ Supplement, pyruvate (Gibco, #31966021)). Tissues were digested in a shaking (100 rpm) 37 °C water bath for 15 min. The obtained cell suspension was pipetted up and down and transferred

onto a 100 μm cell strainer and used for subsequent single-cell sorting. Cells were collected into a 384-well plate format using flow cytometry FACS Aria II BD bioscience). We applied multiple scatter properties to make sure we were sorting for living and single cells.

*Single-cell sequencing.* The SORT-seq procedure was applied as before[2]. Cells were sorted into 384 well plats containing 5μl of VaporLock oil and aqueous solution of 100 nl containing reverse transcriptase (RT) primers, Spike-in RNA molecules, dinucleotide triphosphates (dNTP), and CELL-seq primers which consisted of a 24 bp polyT sequence followed by a 6 bp unique molecular identifier (UMI), a cell-specific barcode, the 5'Illumina TrueSeq2 adapter, and T7 promoter sequence. 384 well plates containing cells and the above-mentioned reagents were incubated for 5 min at 65 °C, after which cDNA libraries were obtained by dispersion of the RT enzyme and second strand mixes with the Nanodrop II liquid handling platform (GC biotech). cDNA libraries in all wells were pooled, followed by separation of the aqueous phase from the oil phase prior to in vitro transcription for linear amplification as performed by overnight incubation at 37 °C. Next Illumina sequencing libraries were prepared using the TruSeqsmall RNA primers (Illumina) and sequenced paired-end at 75-bp read length with Illumina NextSeq. The paired-end reads obtained by Illumina sequencing were mapped with BWA-ALN[3] to the reference genome GRCm38/mm10 downloaded from the UCSC genome browser. We detected on average 24,840 raw unique reads per cell and did not observe any bi- or multimodal distribution of reads per cell, indicating that we indeed sorted exclusively single cells. For quantification of transcripts abundance, the number of transcripts containing unique molecular identifiers per cell-specific barcode was counted for each gene. K-medoids clustering was based on the RaceID2 algorithm to visualize cell clusters using t-distributed stochastic neighbor embedding (t-SNE) and to compute genes up- or downregulated in all cells within the cluster compared with cells not in the cluster[4,5]. Of note, all read counts for mitochondrial genes were discarded due to the high abundance of transcripts coming from these genes in cardiomyocytes, which interfered with downstream clustering.

*Pathway analysis and gene ontology.* The PANTHER Classification System[6] was used in order to categorize the gene expression signature of healthy vs. injured cardiomyocytes based on protein class (Fig. 1). To investigate whether genes shared a similar biological function, we searched for overrepresentation in the Kyoto Encyclopedia of Genes and Genomes (KEGG) pathway and gene ontology (GO) biological processes database using DAVID[7]. Significant enrichment of genes in KEGG and GO terms are shown, p values are corrected for multiple testing using the Benjamini-Hochberg method.

*Heart collection for histological analysis.* Adult hearts were excised from anesthetized animals, washed in cold PBS, and fixed with 4% formalin at room temperature (RT), embedded in paraffin, and sectioned at 4 μm. Paraffin sections were stained with Hematoxylin and Eosin (H&E) for routine histological analysis and Sirius Red (SR) for the detection of collagen according to standard procedures. Slides were visualized using a Zeiss Axioskop 2Plus with an AxioCamHRc and DM4000.

*Human heart samples.* Approval for studies on human tissue samples was obtained from the Medical Ethics Committee of the University Medical Center Utrecht, The Netherlands (12#387). Written informed consent was obtained or in certain cases waived by the ethics committee when obtaining informed consent was not possible due to the death of the patient. None of the co-authors was involved in tissue collection. Tissue samples were anonymized before the access was obtained. In this study, we included tissue from the left ventricular free wall of patients with end-stage heart failure secondary to ischemic heart disease. The end-stage heart failure tissue was obtained during heart transplantation or at autopsy. For immunohistochemistry, samples from three patients were included from which the border zone of the infarcted hearts was used to verify the localization of ZEB2 in hypoxic cells. qPCR analysis for *ZEB2*, *TMSB4*, and *PTMA* was done on 30 tissue samples from the left ventricular free wall of patients with ischemic heart disease (ischemic

region, border zone, and remote region) and 5 samples coming from the left ventricular free wall of the healthy donor.

*Immunohistochemistry.* Immunohistochemistry was performed on paraffin-embedded heart sections. After deparaffinization, rehydration, heat-induced antigen retrieval, and blocking with 0.05% BSA, sections were incubated with primary antibodies overnight at 4 °C. Subsequently, sections were washed and incubated with secondary antibodies for 1 h at RT followed by DAPI 1:5000 (Invitrogen, #D3571) for 10 min at RT. Sections were then washed and mounted with ProLong Gold Antifade Mountant (Invitrogen, #P36934). Antibodies used in this study include mouse anti-ACTN2 (Sigma, #A7732), rabbit anti-ACTN2 (Sigma, #HPA008315), mouse anti-TNNT2 (Abcam, #ab8295), rabbit anti-ZEB2 (Novusbio, #NBP1-77179), rabbit anti-TMSB4 (Immundiagnostik AG, #A9520), goat anti-PTMA (Novusbio, #NBP1-36979) and goat anti-PECAM1 (R&D Systems, #abAF3628). We used a corresponding secondary fluorescent antibody Alexa Fluor 488 donkey anti-mouse IgG (H + L) (Invitrogen, #A21202), Alexa Fluor 488 donkey anti-goat IgG (H + L) (Invitrogen, #A11055), Alexa Fluor donkey anti-rabbit IgG (H + L) (Invitrogen, #A10037), Alexa Fluor 568 donkey anti-mouse IgG (H + L) (Invitrogen, #A21206), Alexa Fluor 568 donkey anti-rabbit IgG (H + L) (Invitrogen, #A10042). FITC-labeled wheat-germ-agglutinin (WGA) (Sigma-Aldrich, #L4895) was used to visualize and quantify cardiomyocyte cross-sectional area with ImageJ software. Blood vessels were stained with PECAM1 and the percentage of the PECAM1 + areas was calculated using ImageJ software.

Immunohistochemistry was also performed on H10 cells transfected with Zeb2 overexpressing vector and siRNA against Zeb2 (as described in detail below). Cells were fixed with 4%PFA, quenched with $NH_4Cl$, permeabilized, blocked with 1% fish gelatin (Gelatin from cold-water fish skin, Sigma-Aldrich, #G7765), and incubated with mouse rabbit anti-ZEB2 antibody (Novusbio, #NBP1-77179) for 25 min at RT prior to incubation with the corresponding secondary Alexa Fluor 488 donkey anti-mouse IgG (H + L) (Invitrogen, #A21202), 20 min at RT. The cells were washed and sealed with a mounting medium (ProLong Gold Antifade Mountant with DAPI, Invitrogen, #P36935). EdU was visualized with the Click-iT EdU Cell Proliferation Imaging Kit, Alexa Fluor 647, ThermoFisher, #C10340, according to the instructions. Images were taken using the Leica TCS SPE confocal microscope.

*Genetic mouse models.* Mice harboring a floxed allele of Zeb2 (Zeb2 fl/fl) were already generated[8] and crossed with mice harboring a Cre recombinase under the control of the murine Myh6 promoter (αMHC-Cre Tg mice)[9] to generate Cre-Zeb2 fl/fl (Zeb2 cKO) mice. Rosa26-LoxStopLoxZeb2[10] were crossed to αMHC-Cre Tg mice to generate αMHC-Cre R26-lslZeb2/lslZeb2 (Zeb2 cTg) mice. Adult mice (8-10 weeks of age) were used for baseline and MI studies. All mice were genotyped by PCR using primers shown in Supplementary Table 7. All animal studies were performed in accordance with local institutional guidelines and regulations. The sample size was determined by a power calculation based upon an echocardiographic effect size. Animal technicians blinded investigators to group allocation during the experiment and when assessing the outcome.

*RNA isolation and quantitative PCR.* Total RNA was isolated from heart ventricles, NRCMs, iPSC-derived cardiomyocytes, H10 cells, or HUVECs using TRIzol reagent according to the manufacturer's instructions. Total RNA (1 μg) was used for mRNA-based reverse transcription using an iScript cDNA Synthesis Kit (Bio-Rad, #1708891). QPCR was performed according to the SYBRgreen based methodology using iQ SYBR Green Supermix (Bio-Rad, #170-8885). Transcript quantities were normalized for endogenous loading. Primer sequences are provided in Supplementary Table 7.

*Western Blot and immunoprecipitation.* Heart tissue lysates were produced in RIPA buffer supplemented with Protease inhibitor cocktail (Roche, #11836170001). Subsequently, samples were boiled in 4x Leammli buffer, including 2% β-mercaptoethanol for 5 min at 99 °C. SDS-PAGE and Western Blotting was performed using Mini-PROTEAN Tetra Vertical Electrophoresis Cell with Mini Trans-Blot (Bio-Rad). Blotted membranes were blocked with 3–5% BSA in TBS-Tween for 30 min at RT. Primary antibody incubation was performed overnight at 4 °C followed by an HRP-conjugated secondary antibody incubation for 1 h at RT. After each antibody incubation, blots were washed 3×15 min with TBS-Tween. Images were visualized using Clarity Western ECL Substrate (Bio-Rad, #170-5061) and Amersham Imager 600. Stripping was performed using Mild stripping buffer (1.5% glycine, 0.1% SDS, 1%Tween pH 2,2). Outputs were normalized for loading. Antibodies used included rabbit anti-BCL-XL (Cell Signaling, #2764), mouse anti-FLAG (Sigma, #F3165), mouse anti-GAPDH (Millipore, #MAB374). Peroxidase-conjugated AffiniPure Rabbit Anti-Mouse IgG (H + L) (Jackson ImmunoResearch, #315-035-003), Peroxidase-conjugated AffiniPure Goat Anti-Rabbit IgG (H + L) (Jackson ImmunoResearch, #111-035-003). Flag-tag pull-down was performed on 1 mg of protein lysate from Zeb2 WT and Zeb2 cTg hearts after MI by using Anti-FLAG Magnetic Beads (Sigma, #M8823). Pull down was performed by rotating the proteins with Anti-FLAG beads at 4 °C overnight. The next day beads were washed and proteins were eluted from beads using trippleFLAG elution buffer (Sigma, #F4799) and used for western blot, as described above. Western blots were quantified by ImageJ.

*Bulk RNA sequencing.* RNA sequencing was performed on heart tissue from Zeb2 fl/fl and Zeb2 cKO mice subjected to MI for 14 days. Total RNA was extracted from the remote zone of hearts using TRIzol reagent as described above. RNA sequencing libraries were prepared from non-ribosomal RNA using the TruSeq Stranded Total RNA Library Prep Kit (Illumina) with Ribo-Zero according to the manufacturer's protocol. Next, strand-specific single-end 75 bp reads were generated on an Illumina NextSeq 500. Reads were aligned and quantified against the Gencode.M4 gtf list for annotated genes using the STAR workflow. Heart libraries were sequenced with a minimum of 14 million reads (16.2 ± 1.9 (mean ± sd)). Differential expression was analyzed using DESeq v1.22[11] using per condition dispersion estimates.

*Isolation of ventricular cardiomyocytes from neonatal rats.* Cardiomyocytes were isolated by enzymatic dissociation of 1-2-day-old neonatal rat hearts. In brief, pups were placed on ice for 5–10 min for light anesthesia. After decapitation, hearts were collected, ventricles were separated from atria and cut into small pieces in a balanced salt solution prior to enzymatic digestion using trypsin (Thermo Fisher Scientific, #15400054) under constant stirring at 37 °C. The supernatant, containing intact cardiomyocytes was collected, centrifuged at 300 g for 4 min, and resuspended in Ham F10 medium (Thermo Fisher Scientific, #11550043) supplemented with 5% FBS, 10% L-glutamine, and 10% Pen-Strep. Collected cells were seeded onto uncoated 100 mm plastic dishes for 1.5 h at 37 °C in 5% $CO_2$ humidified atmosphere. The supernatant, which consists mainly of CMs, was collected, cells were counted and plated on gelatinized 6 well plates 1×10⁶ cells per well. After the 24 h medium was changed to Ham F10 supplemented with Insulin-Transferrin-Sodium Selenite Supplement (Roche), 10% L-glutamine, and 10% Pen-Strep. Cells were used for the hypoxia study and siRNA mediated ZEB2 knock-down.

**Culturing and differentiation of iPSC cells towards cardiomyocytes.** The LUMC0020iCTRL06 human induced pluripotent stem cells (hiPSCs) were used to generate cardiomyocytes using a defined medium with timed addition of growth factors and small molecules[13]. hiPSCs were plated at a density of 50,000 cells/cm² three days prior to differentiation to allow for attachment to Geltrex (Thermofisher Scientific, A1413302) coated cell-culture plates. Differentiation was induced with Cardio Differentiation Medium RPMI 1640 with HEPES and GlutaMax (Life Technologies, #72400021), supplemented with Albumin (human recombinant, Sigma-Aldrich, #A9731), L-Ascorbic Acid 2-phosphate (Sigma-Aldrich, #A8960) with freshly added GSK3B-inhibitor CHIR99021 (Millipore, #361559). On day two, the medium was replaced with Cardio Differentiation Medium supplemented with Wnt-inhibitor IWP2 (Millipore, #681671). On day 4 and 6 the cells received a new differentiation medium without additions. Finally, on day 8 (and every 2/3 days) medium was refreshed with Cardio Culture Medium (RPMI 1640 with HEPES and GlutaMax (Life Technologies, #72400021), supplemented with B27 with Insulin (Life Technologies, #17504044). To enrich for cardiomyocytes, purification was done for 4 days using selection medium (RPMI without D-glucose (Life Technologies, #11879020), supplemented with Albumin, L-Ascorbic Acid 2-phosphate, and Sodium DL-lactate/HEPES buffer (Sigma, #L4263). Differentiation efficiency was determined by flow cytometry for cardiac troponin T (cTNT) expression, using an antibody directed to cTNT (Abcam, #ab45932). Cardiomyocytes were dissociated after selection and reseeded in a density of 400,000 cells/cm² on Geltrex coated cell-culture plates.

*Culturing H10 cells.* H10 cells were cultured in DMEM with GlutaMAX (Thermo Fisher, #10566016) supplemented with 10% FBS (Sigma-Aldrich, #F7524) and 1% Pen-Strep (Thermo Fisher, #15070063). Cells were used for transfection with Zeb2 overexpressing vector and silencing of Zeb2 with siRNA.

*siRNA transfection.* For ZEB2 knock-down, we used Trilencer 27 (Origene, #SR511798). siRNA transfection was performed in neonatal rat cardiomyocytes (NRCMs), H10 cells, and iPSC-derived cardiomyocytes at a final concentration of 10 nM using Lipofectamine 2000 for NRCMs and H10 cells (Thermo Fisher Scientific, #11668027) and Lipofectamine RNAiMAX (Thermo Fisher Scientific, #13778030) for iPSC-derived cardiomyocytes for 24 h. Next, the medium was refreshed for an additional 48 h, and cells were collected for analysis. Cells were subjected to hypoxia (1%$O_2$ and 5%$CO_2$) for 6 h before collection (Supplementary Fig. 6).

*Production of conditioned media.* NRCMs, H10 cells, or iPSC-derived cardiomyocytes were cultured as described above and transfected with siRNA or AAV9 vectors for 24 h after which medium was refreshed and left for an additional 48 h. After that conditioned medium was collected, filtered, and used to treat a new batch of NRCMs and non-cardiomyocytes or HUVECs for 24 h.

*HUVECs culturing and scratch assay.* Human umbilical vein endothelial cells were cultured in EBM-2 BulletKit medium (Lonza, #CC-3162). Scratches in cell monolayers were generated with a 100 μL tip, washed with PBS, and incubated with conditioned media. Cells were imaged at 0, 2, 5, and 20 h with Inverted Routine Microscope ECLIPSE Ts2 (Nikon). The percentage of wound closure was measured by ImageJ by calculating the percentage of area with cells vs total area. For

proliferation assay, HUVECs were incubated with conditioned media for 48 h, EdU treatment took place 24 h before the analysis.

*Spheroid assay.* Human umbilical vein endothelial cells (HUVECs) were trypsinized and resuspended in ECM culture medium (Sciencell, USA) containing 0.6 gr/L methylcellulose (Sigma). Cells were seeded (400 cells per well in 100 μl) in U-bottom 96 well plates and cultured for 24 h at 37 °C and 5% $CO_2$ to allow for the formation of spheroids. The spheroids were collected and resuspended in FBS (Sciencell) containing 2.4 gr/L methylcellulose and mixed 1:1 with collagen I solution containing 3.77 g/L collagen I (Corning, USA), 10% M199 medium (Sigma), 0.018 M HEPES, and 0.2 M NaOH to adjust pH to 7.4. The mixture with the spheroids was allowed to polymerize for 30 min in a 24 well plate. A total of 100 μl of conditioned medium with or without VEGF (50 ng/mL) (Preprotech) was added to the gels. Spheroids were visualized after 24 h using an Olympus IX50 microscope. The cumulative sprout length per spheroid was measured using ImageJ software.

*Mass spectrometry.* H10 cells were plated into T75 flasks and transfected with CMV-control or CMV-Zeb2 vector for 24 h. Medium was refreshed and left for an additional 48 h. Subsequently, cells were collected for molecular analysis and conditioned medium was collected to treat HUVECs and for mass spectrometry. Conditioned medium was separated based on protein size (Centrifugal Filters, Ultracel -3K and -30K, #Amicon Ultra, #UCF800324 and #UCF803024). Mass spectrometry was performed on fraction 3–30kDa. For mass spectrometry; 2 M Urea, 100 mM Tris-HCl, and 10 mM TCEP (end concentration) were added to the supernatants. Samples were incubated at 56 °C for 30 min followed by adding 50 mM CAA and incubation of 5 min at RT. 0.2 μg Trypsin was added and incubated overnight at 37 °C on a shaker. The following day, supernatants were concentrated on zip-tip (C18), washed with 0.1% formic acid, and eluted with 80% acetonitrile/0.1% formic acid. Samples were vacuum concentrated, taken up in 0.1% formic acid, and loaded on a 1.9 μm C18 column coupled to a Proxeon nano-LC system. Samples were measured on the LTQ-Orbitrap Fusion Tribrid; Thermo Fisher Scientific mass spectrometer, using a 200 min gradient and HCD fragmentation. Raw files were analyzed with the Maxquant software version 1.5.2.8 (Cox and Mann, 2008) oxidation of methionine set as variable modifications, and carbamidomethylation of cysteine set as a fixed modification. The Rat protein database of UniProt was searched with both the peptide as well as the protein false discovery rate set to 1%.

*AAV9 delivery to iPSC-derived cardiomyocytes.* Adeno-associated viral vectors (AAV, serotype 9) encoding ZEB2, TMSB4, PTMA, or empty vector were generated in collaboration with Giacca lab in (Trieste, Italy). We used $5×10^3$ viral genome particles per cell to infect iPSC-derived cardiomyocytes.

*Injection of AAV9 vectors.* AAV9 viruses were generated as described above. First, to assess the best way of delivery AAV9-control and AAV9-Zeb2 were injected via three different routes into wild type mice. $0.5 × 10^{11}$ viral genome particles per animal were injected into the myocardium of adult mice (two injections), using a Hamilton syringe. $1×10^{11}$ viral genome particles per animal were injected intravenously into adult mice and intraperitoneally into 2 days old pups using an insulin syringe with a 30-gauge needle. For the injury study, AAV9 intracardiac delivery was performed in C57BL/6 J male mice from Charles River Laboratories (8 weeks of age). Animals received sham or MI surgery as described above. AAV9-control and AAV9-Zeb2 virus at a dose of $0.5 × 10^{11}$ viral genome particles per animal were injected as two intracardiac injections directly after LAD ligation, ($2 × 15$ μl intracardiac injections in an area where the infarct will occur), using a Hamilton syringe (31-gauge, 30° angle). The chest was closed as described above, and animals were placed on 38 °C hot plate for the recovery period. To alleviate pain or discomfort after the surgery, mice were injected with 0.05–0.1 mg/kg of Buprenorphine. Two and four weeks after the surgery and AAV9 injections, cardiac function was analyzed by echocardiography, and hearts were collected for histological and molecular analysis.

*Statistical and reproducibility.* The number of samples (n) used in each experiment is indicated in the legends or shown in the figures and indicates biological replicates. Results are presented as the mean ± standard error of the mean (SEM). Statistical analyses were performed using PRISM (GraphPad Software Inc. version 6). Two groups were statistically compared using the Student's *t*-test. Multiple groups were statistically compared with ordinary one-way ANOVA or two-way ANOVA. Outliers were defined by Grubbs' test (alpha = 0.05). Data are represented as mean ± SEM. Differences were considered statistically significant at $p < 0.05$. In the figures, asterisks indicate statistical significance (*$p < 0.05$, **$p < 0.01$, ***$p < 0.001$, ****$p < 0.0001$) which is also indicated in the individual figures. All representative images of hearts or cells were selected from at least three independent experiments with similar results unless indicated differently in the figure legend.

**Reporting Summary.** Further information on research design is available in the Nature Research Reporting Summary linked to this article.

## Data availability

The authors declare that the main data supporting the findings of this study are available within the article and its Supplementary Information file. All sequencing data that support the findings of this study have been deposited in the National Center for Biotechnology Information Gene Expression Omnibus (GEO) and are accessible through the GEO Series accession number GSE146285 (for SCS data) and GSE151638 (for *Zeb2* cKO RNA-seq data). The mass spectrometry proteomics data have been deposited to the ProteomeXchange Consortium via the PRIDE [1] partner repository with the dataset identifier PXD022212.

Source data are provided with this paper. Extra data are available from the corresponding author upon request. Source data are provided with this paper.

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

## Acknowledgements

We thank Marie-Jose Goumans and Paul Riley for helpful discussions and suggestions. We gratefully acknowledge Hesther de Ruiter, Harry Begthel, Jeroen Korving, Andrea Condi, Maartje Vermunt, Kharishma Patel and Ishan Kallasingh for technical support and Lorena Zentilin for generating the AAV9 vectors.

## Sources of funding

This work was supported by the Leducq Foundation (14CVD04; to E.v.R.), the European Research Council under the European Union's Seventh Framework Programme (ERC Grant Agreement CoG 615708 MICARUS; to E.v.R.) and Belspo IAPVII-07 Devrepair (to D.H. and J.J.H.). D.H. was further supported by Fund for Scientific Research-Flanders (FWO, G.0A31.16). M.M.G. was funded by a Dr. Dekker postdoctoral fellow-ship from the Dutch Heart Foundation (NHS2016T009). H.R.V. was supported by the Proteins@Work initiative of The Netherlands Organization for Scientific Research (NWO) with the grant number: 184.032.201.

## Author contributions

M.M.G., A.K., E.v.R. designed experiments. M.M.G., A.K., D.V., L.K., J.M.-K., V.K., and H.V. performed all experiments. M.M.G., A.K., and B.M. analyzed data. M.M.H.H., J.J. H., D.H., R.A.B, M.G. provided models and materials. M.M.G. and E.v.R. wrote the manuscript.

## Competing interests

The authors declare that they have no competing interests.
