## [Peer Review File · Nature Communications]

Reviewers' comments:

Reviewer #2 (Remarks to the Author):

Gladka and colleagues describe a previously unknown paracrine circuit promoting angiogenesis and tissue repair after MI. The paper is interesting and uses state of the art methodology, but several points need to be addressed:

1) The authors propose that ZEB2 induces TMSB4 and PTMA expression in, and secretion from, cardiomyocytes and that both proteins act as paracrine angiogenic factors after MI. The angiogenic phenotype after MI, however, needs to be better defined: in what region of the infarcted heart does the angiogenic response occur (remote and/or border zone)? What is the time course after MI (PECAM expression was studied only at a single time point)? Are the newly formed capillaries perfused (i.e. functional)? This could be studied in mice injected, for example, with labeled isolectin B4.

2) The authors report that also BCL-XL is induced in a ZEB2-dependent manner. This needs to be further explored. Is BCL-XL induced in cardiomyocytes? Is ZEB2 directly regulating BCL-XL? Does this have an influence on post MI cardiomyocyte survival (apoptosis), e.g. in the border zone?

3) Infarct scar formation in ZEB2-deleted or -overexpressing mice needs to be better described (e.g. scar size in [%] of LV circumference, scar thickness, rupture rates, impact on mortality).

4) Crucial experiments are performed in H10 cells (not sure if these are truly cardiomyocytes), neonatal rat ventricular cardiomyocytes, and human iPSC-derived cardiomyocytes. Why did the authors not focus on a single cell type? This needs a comment. I would like to see some data proving that the used 'iPSC-derived cardiomyocytes' are truly cardiomyocytes.

5) Although TMSB4 and PTMA have been described as secreted proteins, the authors need to show that both factors are actually secreted from ZEB2 expressing cardiomyocytes (TMSB4 may also act intracellularly). Can these factors be detected by ELISA (or else) in the supernatant of ZEB2 expressing cardiomyocytes? Can these proteins be detected in the circulation (blood) of ZEB2 overexpressing mice? The authors show that conditioned supernatants from ZEB2 expressing cardiomyocytes promote angiogenesis. Do they have neutralizing antibodies at hand (or can they use another experimental setup) to demonstrate that this is a TMSB4 and/or PTMA-dependent effect?

Additional points:

a) The authors propose that several secreted proteins (Fig. 3I) are induced by ZEB2 in cardiomyocytes. Have these factors also been identified as being expressed/upregulated after MI in cardiomyocytes in the initial single cell sequencing exercise? Please comment.

b) Results from many experiments are presented simultaneously in the main manuscript (Figures 1-7) and in the Supplement (Suppl. Figures and Tables) – e.g. MI data are presented in the manuscript but sham data in the supplement or day 28 data are presented in the manuscript but day 14 data in the supplement. Oftentimes, this is not well explained in the text and makes the manuscript tedious to read. Consider moving entire data sections to the supplement rather than tearing the data apart (as is done now).

c) The discussion is rather superficial and largely recapitulates the results. The authors may want to provide more info on TMSB4 and PTMA. Are these genes related? Sequence similarities? What is known from previous studies regarding their tissue / cellular expression patterns and role(s) in tissue/heart angiogenesis under normal conditions or after stress?

d) Lines 81-82: this manuscript uses a mouse acute MI model; not sure how results from this study should then inform ‘heart failure therapies’?

e) Line 180: effects on ‘endothelial genes’ are not statistically significant.

f) Line 206: a ‘trend’ usually implies a P value <0.10. These passages need to be reworded.

g) Line 274: the AAV9 vector used to overexpress ZEB2 utilizes a CMV promotor and does not deliver ZEB2 specifically to cardiomyocytes (Suppl. Fig. 8). This should be noted as a limitation.

h) Suppl. Fig 5: there appear to be 38 or 40 ZEB2 motifs in the mouse and human PTMA genes, which seems quite a lot. What is the ZEB2 consensus motif?

Reviewer #3 (Remarks to the Author):

In this manuscript, the authors describe the identification of a transcription factor, ZEB2, as a key factor to control the damage response of cardiomyocytes. The authors conducted a series of single cell RNA seq analysis and found that ZEB2 is increased in stressed cardiomyocytes. The activation of ZEB2 further induces a cross talk between cardiomyocytes and endothelial cells to protect cardiomyocytes by enhancing angiogenesis. To validate this hypothesis, the authors conducted a gain-of-function and a loss-of-function assays of ZEB2 gene and carefully validated the ZEB2 is truly a key factor. To further identify the effector proteins downstream to ZEB2, the authors employed a mass-spectrometry assay and identified TMSB4 and PTMA as main paracrine protein factors. Those proteins are released from cardiomyocytes to stimulate angiogenesis by enhancing cellular migration of surrounding endothelial cells. Delivery of ZEB2 to the infarcted heart, which mimics a therapeutic intervention of the disease, resulted in enhanced angiogenesis and prevention of cardiac dysfunctions, which shows the obtained knowledge is potentially applicable for the treatment of patients.

Overall, I consider the manuscript should convey various novel insights, which would lead to better understandings of cardiac cellular networks and future development of a novel therapeutic method

towards repairing and the restoring the hear functions after the heart attack. However, I'm not yet fully convinced that how downstream factors, TMSB4 and PTMA4, invoke the response of epithelial cells, which eventually leads to improved phenotypic appearance. The followings are the points which I believe should be needed to confirm the claims of the manuscript.

Major points:

1. Difference in the phenotypic appearance between the wild type and transgenic models, as observed by echo (as LVIDs or FS), seems not always large, even though statistically significant. On the other hand, the results of the molecular analyses, such as the induction of TMSB4 and PTM4 by ZEB2, seems much clearer. I wonder if the paracrine model of these factors as describe here would truly bring about reduced pathogenic remodeling in patients after all.

2. More or less, the presented data which is directly associated with TMSB4 and PTM4 is from the wound healing assays in vitro. In addition to the indirect role of ZEB2, further careful validations should be included before the authors conclude that TMSB4 or PTMA are responsive factors for enhanced angiogenesis, and thereby reducing the pathological remodeling.

Minor points:

3. Further details should be analyzed (or at least discussed) on:

-How ZEB2 regulates the target genes, TMSB4 and PTMA

-How ZEB2 itself is regulated in response to heart stress

-How the PTMA and TMSB4 signals, either collectively or independently, invokes the changes in endothelial cells, and triggers downstream events.

4. References are generally old on the previous knowledge of TMBS4 and PTM4. I wonder if there has been any update since then, especially their involvement with the cellular migrations or angiogenesis?

5. I wonder how diverse the ZEB2 expression and its response, depending on each individual cell, is in a healthy condition and a stressed condition in wild type and knockout mice. Single cell analysis for the transgenic models or therapeutic delivery models would address this issue. It is also important to observe the diversity of the response in terms of the region of the response within the tissue.

6. It would be also interesting to examine the physiological outcomes of the possible therapeutic delivery of TMSB4 and PTMA as well as ZEB2.

7. The results of the single cell RNA seq analysis in Figure 1 remain unattended except for cardio myocyte. It would be interesting to further analyze the responses of other cell types, such as fibroblast cells and other cells, which collectively constitute the cardio-vascular cellular system.

8. I'm not sure whether it is allowed to leave all the method descriptions to supplementary section.

9. Unless additional evidence is provided, Figure 8 should be toned down. Items shown in the bottom margin are not always fully proven.

10. "MI" should be spelled out at its first appearance in the abstract (line 40).

Possible typos: 1-Pearson (line p.97); Immunofluorescence-> Immunostaining assay (line 117).

Reviewer #4 (Remarks to the Author):

This manuscript by Gladka et al describes a potential novel role for Zeb2 in regulating myocardial function post-myocardial infarction via its control of the secretion of angiogenic molecules. Overall, the manuscript was quite well written and the data clearly presented. However, there are a few areas that need to be clarified.

1. If these two new paracrine factors are essential for mediating Zeb2 effects, an antibody neutralization of the conditioned medium would demonstrate that they are critical for mediating Zeb2's pro-angiogenic effect. Also what is the nature of Zeb2 regulation of these genes? Are there Zeb2 binding sites in their promoters for example?

2. The scratch wound healing assay is a read out of both effects on cell migration and cellular proliferation. As such it should be complemented with more clear migration assays such as the

Boyden chamber assays. As well, the effect on tube formation is a well accepted in vitro readout of angiogenesis and its addition to the current study would strengthen the manuscript.

3. Cardiac function is shown for the fl/fl mice versus control after MI but should be also shown prior to MI to show if there is any effect of strain/deletion on basal cardiac parameters.

4. In addition to fractional shortening, ejection fraction is a critical clinical parameter of cardiac function and should be measured. The authors may also want to look at E/A ration which is a measure of diastolic function.

5. Was cardiac fibrosis altered in this model? It should be discussed as it is known to affect cardiac function.

6. The cardiac specific deletion reduced total cardiac Zeb2 by approximately 50% in the post-MI heart. What non-cardiomyocyte cells express Zeb2 in this setting?

7. Cell death should be measured in addition to looking at levels of Bcl-XL

8. The phrase “trending decrease” should be removed from the results and the results more clearly describe- no significant change. The possible trends could be brought up in the discussion.

9. Acronyms should be spelled out in the abstract.

"Cardiomyocytes stimulate angiogenesis after ischemic injury in a ZEB2-dependent manner"

We are grateful to the editors and reviewers for their comments and we tried to address all issues raised point-by-point. We feel that by doing so we were able to significantly improve the quality of the manuscript.

Reviewers' comments:

Reviewer #2 (Remarks to the Author):

Gladka and colleagues describe a previously unknown paracrine circuit promoting angiogenesis and tissue repair after MI. The paper is interesting and uses state of the art methodology, but several points need to be addressed:

We would like to thank the reviewer for the positive feedback and constructive comments. We have attempted to address all issues raised by this reviewer to improve the quality of our work.

1) The authors propose that ZEB2 induces TMSB4 and PTMA expression in, and secretion from, cardiomyocytes and that both proteins act as paracrine angiogenic factors after MI. The angiogenic phenotype after MI, however, needs to be better defined: in what region of the infarcted heart does the angiogenic response occur (remote and/or border zone)? What is the time course after MI (PECAM expression was studied only at a single time point)? Are the newly formed capillaries perfused (i.e. functional)? This could be studied in mice injected, for example, with labeled isolectin B4.

To answer what the localization and timing of the angiogenic response are, we evaluated mRNA expression of numerous angiogenic/endothelial markers in our *in vivo* gain and loss-of-function mouse models in both the remote and infarct/border zone area.

We observed that the angiogenic response mainly occurs in the remote area of the injured heart (Figure 1).

Per the reviewer's suggestion, we also checked the expression of *Pecam1* in cardiac samples from both the infarcted/border zone as well as the remote area of the injured heart at different time points following ischemic injury (**Figure 2**). We observed an initial increase in the expression in the infarcted/border zone region, which was decreased at 14 days post-IR. There were no detectable differences in the *Pecam1* expression in the remote area.

Additionally, we planned to test the functionality of newly formed capillaries by performing isolectin b4 injections. Unfortunately, because of the ongoing quarantine, we were only able to perform the experiment in the *Zeb2* cKO study. These data indicated that ZEB2 deletion from cardiomyocytes resulted in a decrease in perfused capillaries in the heart post-MI (**Figure 3a-b**), which corresponds to a decline in PECAM1 positive areas in those hearts (**Figure 3c-d**). Based on these observations we predict that the opposite effect will be observed in the *Zeb2* cTg mice (which we are not able to perform at the moment) as there is an increase in PECAM1 positive areas (**Figure 3e-f**).

2) The authors report that also BCL-XL is induced in a ZEB2-dependent manner. This need to be further explored. Is BCL-XL induced in cardiomyocytes? Is ZEB2 directly regulating BCL-XL? Does this have an influence on post MI cardiomyocyte survival (apoptosis), e.g. in the border zone?

To answer whether BCL-XL is directly regulated by ZEB2 in cardiomyocytes, we treated iPS-derived cardiomyocytes with an adenovirus overexpressing ZEB2. We were unable to detect an increase in BCL-XL, indicating that ZEB2 does not directly regulate BCL-XL in healthy cardiomyocytes (**Figure 4**). As BCL-XL is induced

in the injured hearts upon ZEB2 overexpression, where we see an increase in survival, this is most likely due to a secondary effect of enhanced ZEB2-dependent angiogenesis which protect cardiomyocytes from apoptosis.

3) Infarct scar formation in ZEB2-deleted or -overexpressing mice needs to be better described (e.g. scar size in [%] of LV circumference, scar thickness, rupture rates, impact on mortality).

We used echocardiography data to measure scar size by assessing the percentage of LV circumference in B-mode at long axes view. **Figures 5a** and **5b** show representative echocardiographic images of hearts from *Zeb2* cKO MI and *Zeb2* cTg MI studies, respectively. The percentage of infarct size in both studies was quantified (**Figure 5c-d**). We observed increased infarct size in *Zeb2* cKO mice compared to *Zeb2* fl/fl controls, which corresponds with a decline in FS in *Zeb2* cKO mice (**Figure 5e**). On the contrary, infarct size was smaller in *Zeb2* cTg mice compared to wild type controls, which was in line with a better cardiac function, as shown by improved FS (**Figure 5f**).

We looked at the survival of animals post-MI in both groups and observed an increase in mortality of *Zeb2* cKO mice (**Figure 5g**). There was no apparent difference in survival in the *Zeb2* cTg study (**Figure 5h**). Important to note is that both studies were performed by different animal surgeons, resulting in differences in the severity of MI. The MI in *Zeb2* cKO study was milder, whereas it was quite severe in the *Zeb2* cTg study. This was also mentioned in the methods section of the original manuscript. We analyzed both studies separately and we did not make any comparisons between both studies.

4) Crucial experiments are performed in H10 cells (not sure if these are truly cardiomyocytes), neonatal rat ventricular cardiomyocytes, and human iPSC-derived cardiomyocytes. Why did the authors not focus on a single cell type? This needs a comment. I would like to see some data proving that the used 'iPSC-derived cardiomyocytes' are truly cardiomyocytes.

We indeed used multiple *in vitro* models. We used H10 cells for mass spectrometry experiments since we needed a substantial number of cells from a pure population to produce enough medium for the analysis. For our validation experiments we decided to use cells that more closely resemble human adult cardiomyocytes, meaning neonatal rat ventricular cardiomyocytes and iPSC-derived cardiomyocytes.

The purity of our iPSC-derived cardiomyocytes was between 90-95%, which is also shown in **Figure 6**, where cells were stained with cardiac troponin T (red) and DAPI (blue).

Figure 6

5) Although TMSB4 and PTMA have been described as secreted proteins, the authors need to show that both factors are actually secreted from ZEB2 expressing cardiomyocytes (TMSB4 may also act intracellularly). Can these factors be detected by ELISA (or else) in the supernatant of ZEB2 expressing cardiomyocytes? Can these proteins be detected in the circulation (blood) of ZEB2 overexpressing mice? The authors show that conditioned supernatants from ZEB2 expressing cardiomyocytes promote angiogenesis. Do they have neutralizing antibodies at hand (or can they use another experimental setup) to demonstrate that this is a TMSB4 and/or PTMA-dependent effect?

We identified PTMA and TMSB4 by mass spectrometry that we performed on the conditioned medium from ZEB2 overexpressing cardiomyocytes. We attempted to perform WB (with several commercially available antibodies) on plasma from our *in vivo* studies as well as on the conditioned medium from ZEB2-overexpressing cardiomyocytes but were unable to detect a reliable signal. As we are currently unable to generate additional; custom-made antibodies, we instead focused on the downstream pro-angiogenic function of those factors to prove they are secreted by cardiomyocytes and at least partially responsible for the effect on angiogenesis.

To show this, we performed a spheroid-based sprouting assay, which is a well-established method to study capillary-like tube formation of cultured endothelial cells. The advantage of this assay is that it provides the possibility to study angiogenesis in a 3D environment. We treated iPSC-derived cardiomyocytes with siRNA-control, siRNA-ZEB2, siRNA-TMSB4, siRNA-PTMA and combination of siRNA-TMSB4 and siRNA-PTMA (**Figure 7a**). We collected conditioned medium from treated cardiomyocytes and used it to treat HUVEC cells in sprouting assay (**Figure 7a**). We imaged multiple spheroids (**Figure 7b**) and quantified a total number of sprouts per spheroid as well as cumulative sprout length per spheroid. We observed that ZEB2 inhibition resulted in a significant decrease in the number of sprouts per spheroid (**Figure 7c**) and a non-significant decrease in cumulative sprout length per spheroid (**Figure 7d**). Inhibition of TMSB4 and PTMA separately did not affect the number and length of sprouts, but when inhibited together, they showed similar results as ZEB2 inhibition.

Figure 7
We also check the effect of inhibition of ZEB2, TMSB4, PTMA4 and TMSB4 + PTMA together in cardiomyocytes (as described in **Figure 7a**) on endothelial cell proliferation. We treated HUVEC with conditioned medium and analyzed 48 hours later (**Figure 8a**). We analyzed EdU positive HUVEC cells and observed a significant decrease in EdU positive nuclei when treated with conditioned medium from ZEB2 inhibited cardiomyocytes (**Figure 8b-c**). Even though we did not reach significance in the decrease of EdU positive HUVEC cells in TMSB4 and PTMA treated conditions, we observed a very pronounced decrease in the expression of cell cycle genes in all conditions (**Figure 8d-f**). Additionally, we observed a decrease in *Pecam1*, *Angiogenin* and *Endoglin*, which are involved in angiogenesis and cell migration (**Figure 8g-i**). Together we showed that inhibition of ZEB2 in cardiomyocytes (and to a lower extent the individual factors TMSB4 and PTMA), reduces endothelial cell sprouting, proliferation and contributes to inhibition of angiogenic response. **Figure 7 and 8 have now also been included in the manuscript.**

Figure 8

Additional points:

a) The authors propose that several secreted proteins (Fig. 3I) are induced by ZEB2 in cardiomyocytes. Have these factors also been identified as being expressed/upregulated after MI in cardiomyocytes in the initial single cell sequencing exercise? Please comment.

Yes, these factors were also identified in our original single-cell sequencing data (Figure 3I in the original manuscript). They were all upregulated in cardiomyocytes post-MI when compared to sham cardiomyocytes (Figure 9).

Figure 9

b) Results from many experiments are presented simultaneously in the main manuscript (Figures 1-7) and in the Supplement (Suppl. Figures and Tables) – e.g. MI data are presented in the manuscript but sham data in the supplement or day 28 data are presented in the manuscript but day 14 data in the supplement. Oftentimes, this is not well explained in the text and makes the manuscript tedious to read. Consider moving entire data sections to the supplement rather than tearing the data apart (as is done now).

We now attempted to be more clear throughout the text.

c) The discussion is rather superficial and largely recapitulates the results. The authors may want to provide more info on TMSB4 and PTMA. Are these genes related? Sequence similarities? What is known from previous studies regarding their tissue / cellular expression patterns and role(s) in tissue/heart angiogenesis under normal conditions or after stress?

The discussion was rewritten according to the reviewer's suggestions.

d) Lines 81-82: this manuscript uses a mouse acute MI model; not sure how results from this study should then inform 'heart failure therapies'?

This is, of course, a long-standing goal that requires additional research. However, we believe that the data provided in our manuscript may hold great promise for future ischemia-induced heart failure therapies. We have now adapted this sentence.

e) Line 180: effects on 'endothelial genes' are not statistically significant.

The reviewer is correct that there is no significant decrease in endothelial markers, however, there is a definite trend (n=3). This might be due to a low number of endothelial cells in the non-cardiomyocyte fraction, which consist mainly of fibroblasts. However, we do think this data is relevant as the purpose of this experiment was to show that the conditioned medium affects mainly non-myocytes.

f) Line 206: a 'trend' usually implies a P value <0.10. These passages need to be reworded.

We adjusted the text accordingly.

g) Line 274: the AAV9 vector used to overexpress ZEB2 utilizes a CMV promoter and does not deliver ZEB2 specifically to cardiomyocytes (Suppl. Fig. 8). This should be noted as a limitation.

AAV serotype 9 vectors are commonly used for the gene transfer to cardiomyocytes. AAV9 shows robust cardiac transduction also without a cardiac-specific promoter due to preferential transduction of non-dividing cells like cardiomyocytes. However, we agree with the reviewer that for even better specificity, the use of cardiac promoters would be recommended. We indicated this now in the text.

h) Suppl. Fig 5: there appear to be 38 or 40 ZEB2 motifs in the mouse and human PTMA genes, which seems quite a lot. What is the ZEB2 consensus motif?

Indeed, based on oPOSSUM (which is a web-based system for the detection of over-represented conserved transcription factor binding sites), there are many predicted binding sites of ZEB2 in the mouse and human

promoters of *PTMA* and *TMSB4*. We screened 10 kb upstream and 10 kb downstream from the transcription starting site (TSS) of *PTMA* and *TMSB4* with the following settings: Conservation cutoff: 0.40 and Matrix score threshold: 85%.

ZEB2 is binding to the following E-box motif:

ZEB2 E-box -CACCT- -GTGGA-

Reviewer #3 (Remarks to the Author):

In this manuscript, the authors describe the identification of a transcription factor, ZEB2, as a key factor to control the damage response of cardiomyocytes. The authors conducted a series of single cell RNA seq analysis and found that ZEB2 is increased in stressed cardiomyocytes. The activation of ZEB2 further induces a cross talk between cardiomyocytes and endothelial cells to protect cardiomyocytes by enhancing angiogenesis. To validate this hypothesis, the authors conducted a gain-of-function and a loss-of-function assays of ZEB2 gene and carefully validated the ZEB2 is truly a key factor. To further identify the effector proteins downstream to ZEB2, the authors employed a mass-spectrometry assay and identified TMSB4 and PTMA as main paracrine protein factors. Those proteins are released from cardiomyocytes to stimulate angiogenesis by enhancing cellular migration of surrounding endothelial cells. Delivery of ZEB2 to the infarcted heart, which mimics a therapeutic intervention of the disease, resulted in enhanced angiogenesis and prevention of cardiac dysfunctions, which shows the obtained knowledge is potentially applicable for the treatment of patients.

Overall, I consider the manuscript should convey various novel insights, which would lead to better understandings of cardiac cellular networks and future development of a novel therapeutic method towards repairing and the restoring the hear functions after the heart attack. However, I'm not yet fully convinced that how downstream factors, TMSB4 and PTMA4, invoke the response of epithelial cells, which eventually leads to improved phenotypic appearance. The followings are the points which I believe should be needed to confirm the claims of the manuscript.

We thank the reviewer for the comments. We have now added in additional experimental data which we think further strengthen our story.

Major points:

1. Difference in the phenotypic appearance between the wild type and transgenic models, as observed by echo (as LVIDs or FS), seems not always large, even though statistically significant. On the other hand, the results of the molecular analyses, such as the induction of TMSB4 and PTM4 by ZEB2, seems much clearer. I wonder if the paracrine model of these factors as describe here would truly bring about reduced pathogenic remodeling in patients after all.

The functional effect of ZEB2 overexpression *in vivo* results in substantial improvement in cardiac function (**Figure 10a-e**). Additionally, left ventricular (LV) mass and heart weight to body weight ratio (HW/BW) ratio were decreased, indicating reduced pathological remodeling induced by ischemic injury (**Figure 10f-g**). We used echocardiography data to measure scar size by assessing the percentage of LV circumference in B-mode at long axes view. **Figure 10h** shows representative echocardiographic images of hearts from Zeb2 cTg MI study. Using echocardiographic images, we observed a smaller infarct size in Zeb2 cTg mice compared to wild type controls (**Figure 10i**), which was in line with an improved cardiac function in the Zeb2 Tg after MI (**Figure 10a-e**). We believe the effects of ZEB2 overexpression will translate into a beneficial outcome in patients suffering from ischemic heart disease by reducing infarct size and secondary remodeling which, will subsequently improve cardiac function.

Figure 10

2. More or less, the presented data which is directly associated with TMSB4 and PTM4 is from the wound healing assays in vitro. In addition to the indirect role of ZEB2, further careful validations should be included before the authors conclude that TMSB4 or PTMA are responsive factors for enhanced angiogenesis, and thereby reducing the pathological remodeling.

Multiple studies so far have implicated TMSB4 and PTMA in the cardiac angiogenic response. The group of Paul Riley previously described a critical role for TMSB4 in coronary vessel development and neovascularization (Smart et al. 2007). In another publication by Ziegler et al. from 2017, it was also shown that TMSB4 induces angiogenic sprouting and vascular maturation. Those are only two of many publications confirming the role of TMSB4 as a pro-angiogenic factor. Regarding the function of PTMA, it was already described by Malinda et al. in 1998 that PTMA (known as Thymosin alpha1) stimulates endothelial cell migration, angiogenesis and wound healing.

Here we demonstrate that the observed angiogenic phenotype is ZEB2-dependent and that this pro-angiogenic function of ZEB2 occurs via TMSB4 and PTMA regulation and secretion from cardiomyocytes. We did so by using a spheroid-based sprouting assay, which is a well-established method to study capillary-like tube formation of cultured endothelial cells. The advantage of this assay is the possibility to study angiogenesis in a 3D environment. We treated iPSC-derived cardiomyocytes with siRNA-control, siRNA-ZEB2, siRNA-TMSB4, siRNA-PTMA and combination of siRNA-TMSB4 and siRNA-PTMA (Figure 11a). We collected conditioned medium from treated cardiomyocytes and used it to treat HUVEC cells in sprouting assays (Figure 11a). We imaged multiple spheroids (Figure 11b) and quantified a total number of sprouts per spheroid as well as cumulative sprout length per spheroid. We observed that ZEB2 inhibition resulted in a significant decrease in the number of sprouts per spheroid (Figure 11c) and a non-significant decrease in cumulative sprout length per spheroid (Figure 11d). Inhibition of TMSB4 and PTMA separately did not affect the number and length of sprouts, but when inhibited together, they show similar results as ZEB2 inhibition. These data show that ZEB2 is required for sprout formation. As inhibition of TMSB4 and PTMA show a similar pattern as ZEB2 alone, we conclude these factors

to be involved in the angiogenic response. We also performed the same experiment where we looked at the sprout formation of endothelial cells when treated with conditioned medium from cardiomyocytes treated with AAV9-ZEB2, AAV9-TMSB4, AAV9-PTMA and a combination of AAV9-TMSB4 and AAV9-PTMA (**Figure 12a**). VEGF was used to stimulate sprout formation. We imaged multiple spheroids (**Figure 12b**) and quantified a total number of sprouts per spheroid as well as cumulative sprout length per spheroid. We observed an increase of a total number of sprouts per spheroids and cumulative sprout length when treated with the conditioned medium from ZEB2 overexpressing cardiomyocytes under VEGF-stimulated conditions (**Figure 12c-d**). Additionally, we observed that TMSB4 alone could trigger the increase in the number and total length of sprouts (**Figure 12c-d**). Both experiments indicate a more potent effect of ZEB2 on angiogenesis than TMSB4 and PTMA alone or even together.

In Figure 5 of the original manuscript, we showed that inhibition of ZEB2 in the conditioned medium from cardiomyocytes slowed down wound healing (scratch assay in HUVEC cells). We could partially rescue this effect by overexpressing TMSB4 and PTMA. This indicates that TMSB4 and PTMA alone are capable of triggering pro-angiogenic effects even in the absence of ZEB2. These data are shown again below (**Figure 13a-b**).

We now also generated data where we studied the effect of ZEB2, TMSB4, PTMA4 and TMSB4 + PTMA inhibition and overexpression in cardiomyocytes on endothelial cell proliferation (**Figure 14**). To do so, we treated iPSC-derived cardiomyocytes with siRNA-control, siRNA-ZEB2, siRNA-TMSB4, siRNA-PTMA and combination of siRNA-TMSB4 and siRNA-PTMA. We collected conditioned medium from treated cardiomyocytes and used it to treat HUVEC cells for 48 hours (**Figure 14a**). We analyzed EdU positive HUVEC cells and observed a significant decrease in EdU positive nuclei when treated with conditioned medium from ZEB2 inhibited cardiomyocytes (**Figure 14b-c**). Even though we did not reach significance in the decrease of EdU positive HUVEC cells in TMSB4 and PTMA treated conditions, we observed a very pronounced decrease in the expression of cell cycle genes in all conditions (**Figure 14d-f**). Conditioned medium from cardiomyocytes overexpressing ZEB2, TMSB4, PTMA and TMSB4 + PTMA together induced a significant increase in EdU positive nuclei when treated with all conditions on endothelial cell proliferation (**Figure 14g**). HUVECs treated with conditioned medium and analyzed 48 hours later showed a significant increase in EdU positive nuclei when treated with all conditions (**Figure 14g-i**). Together this shows that inhibition and overexpression of ZEB2 in cardiomyocytes (and to the lower extent individual factors TMSB4 and PTMA) influences endothelial cell sprouting, proliferation and contribute to inhibition of the angiogenic response.

Figure 12

Figure 13

Figure 14

Minor points:

3. Further details should be analyzed (or at least discussed) on:

- How ZEB2 regulates the target genes, TMSB4 and PTMA

Based on oPOSSUM (which is a web-based system for the detection of over-represented conserved transcription factor binding sites), there are many predicted binding sites of ZEB2 in the mouse and human promoters of *Tmsb4* and *Tmsb4*. We screened 10 kb upstream and 10 kb downstream from the transcription starting site (TSS) of *Ptma* and *Tmsb4* with the following settings: Conservation cutoff: 0.40 and Matrix score threshold: 85%. As shown in Supplementary Figure 5J we identified multiple potential binding sites.

Additionally, we made use of existing Chip-seq dataset <https://www.encodeproject.org/targets/ZEB2-human/> and identified ZEB2 binding motifs in enhancer regions of both TMSB4 (Figure 15a) and PTMA (Figure 15b), indicating that the ZEB2 protein directly binds to those enhancers causing an increase in TMSB4 and PTMA expression.

Figure 15
- How ZEB2 itself is regulated in response to heart stress

As a part of another project, we are investigating how ZEB2 is regulated in response to ischemic injury. We revealed that *Zeb2* is a target gene of HIF1 α , a transcription factor that is regulated by hypoxia. We applied the Tomo-seq technique (Lacruz et al. 2017) to identify novel mechanisms that underlie cardiac remodeling after ischemic injury. RNA sequencing on consecutive sections of an infarcted mouse heart allowed us to obtain a genome-wide gene expression signature with high spatial resolution going from the infarct to the remote area (**Figure 16a-b**). We identify that the spatial expression of *Zeb2* is in line with the expression of *Hif1 α* (**Figure 16c-e**). Extensive *in vivo* studies enabled to show that there is a robust HIF1 α -dependent increase in *Zeb2* expression in the hypoxic hearts in the infarcted area following ischemic damage, as shown by gene expression (**Figure 16f**). To address the possible regulation of ZEB2 by HIF1 α in cardiomyocytes, we induced targeted knockdown of endogenous Hif1 α by a specific siRNA in cardiomyocytes subjected to normoxia or hypoxia for 6h. After treatment with siRNA against Hif1 α , we observed a significant decrease not only of *Hif1 α* and known *Hif1 α* target genes but also of *Zeb2* transcripts levels (**Figure 16g-h**). This might be due to multiple conserved hypoxia-responsive elements (HRE) in the *Zeb2* promoter region (**Figure 16i**). Moreover, we observed a positive correlation of *Hif1 α* and *Zeb2* in single cardiomyocytes isolated from the injured heart (**Figure 16j**) as well as a positive correlation of both factors in human ischemic hearts. These data are strongly suggesting a link between those two factors whereby HIF1 α promotes the expression of ZEB2 during hypoxia-induced cardiac injury.

We are currently pursuing this link in an additional study.

Figure 16

Figure 16. Zeb2 is a downstream factor of Hif1α in hypoxic cardiomyocytes. (a) Study design. **(b)** The indicated area was collected for sectioning and subsequent RNA sequencing (tomo-seq) **(c)** Tomo-seq expression patterns of Hif1α and 10 co-expressed genes in heart tissue 14 days post-IR. **(d)** Gene ontology analysis of 100 genes co-expressed with Hif1α in the ischemic heart. **(e)** Tomo-seq expression patterns of Hif1α and Zeb2 in heart tissue 14 days post-IR. **(f)** qPCR analysis of Hif1α and Zeb2 in post-MI hearts. **(g)** qPCR analysis of Hif1α and **(h)** Zeb2 in NRCM treated with control or a Hif1α siRNA during normoxia or hypoxia. **(i)** Comparison of the Zeb2 genomic regions between mouse and human as conservation percentage of 10.0 kb genomic region upstream of the Zeb2 gene **(j)** Single-cell sequencing analysis of

mouse injured cardiomyocytes showing positive correlation between cells expressing both Hif1α and Zeb2 as shown by normalized read counts **(k)** Pearson correlation between the expression of HIF1α and ZEB2 determined by qPCR analysis on human cardiac tissue.

- How the PTMA and TMSB4 signals, either collectively or independently, invokes the changes in endothelial cells, and triggers downstream events.

It was previously described in extensive detail about the prominent function of TMSB4 and PTMA in endothelial cells. Earlier studies indicated that TMSB4 plays an essential role in facilitating cardiac neovascularization, promoting cell proliferation and differentiation and maintaining myocardial function following ischemic injury (Smart et al. 2007, Ziegler et al. 2017, Lv et al. 2013, Philip et al. 2003, Grant et al. 1999). PTMA was implicated in endothelial cell migration, angiogenesis and wound healing (Malinda et al. 1998)

Both peptides were patented for the development of wound healing and pro-angiogenic therapies.

4. References are generally old on the previous knowledge of TMSB4 and PTMA. I wonder if there has been any update since then, especially their involvement with the cellular migrations or angiogenesis?

We have attempted to update the references where applicable.

5. I wonder how diverse the ZEB2 expression and its response, depending on each individual cell, is in a healthy condition and a stressed condition in wild type and knockout mice. Single cell analysis for the transgenic models

or therapeutic delivery models would address this issue. It is also important to observe the diversity of the response in terms of the region of the response within the tissue.

One of the main limitations of single-cell sequencing techniques is the low detection efficiency of transcription factors and genes in general. While we can sometimes still detect transcription factors, the absence of the expression by the single-cell sequencing method does not necessarily mean that they are not expressed. For this reason, it will be challenging to examine ZEB2 effects purely based on this readout.

We can however clearly observe the diversity of the response in different regions in the heart due to Zeb2 gain and loss-of-function by bulk RNA sequencing, which indicates that the angiogenic response mainly occurs in the remote area of the injured heart (**Figure 17**)

6. It would be also interesting to examine the physiological outcomes of the possible therapeutic delivery of TMSB4 and PTMA as well as ZEB2.

As a part of another study that we are carrying out in our lab, we tested the potential therapeutic effect of the delivery of TMSB4 and PTMA in the injured heart. We induced sham or IR (ischemia/reperfusion) and therapeutically injected AAV9-control or AAV9-Tmsb4 together with AAV9-Ptma for two weeks (**Figure 18a**). Echocardiographic analysis showed an improved cardiac function upon delivery of Tmsb4 and Ptma. Specifically, we saw a decrease in relaxation time (measured by IVRT), a higher ejection fraction (EF), and less remodeling as indicated by the left ventricular posterior wall (LVPW) thickness (**Figure 18b**). There was also an increase in the majority of tested pro-angiogenic markers, confirming a potential therapeutic benefit of Tmsb4 and Ptma overexpression after ischemic injury (**Figure 18c-d**). We are currently pursuing this further.

Figure 18

7. The results of the single cell RNA seq analysis in Figure 1 remain unattended except for cardio myocyte. It would be interesting to further analyze the responses of other cell types, such as fibroblast cells and other cells, which collectively constitute the cardio-vascular cellular system.

We agree with the reviewer, and in fact, we already performed such a study (Gladka et al. Circulation 2018). We performed single-cell sequencing in hearts subjected to ischemia/reperfusion (IR). We characterized the differences in gene expression in all main cardiac cell types. We found disease enriched or even disease-specific subpopulations. For this study, we wanted to focus on identifying a transcription factor that is enhanced in injured cardiomyocytes and study its function.

8. I'm not sure whether it is allowed to leave all the method descriptions to supplementary section.

We have now adjusted it in the manuscript according to the guidelines of the journal.

9. Unless additional evidence is provided, Figure 8 should be toned down. Items shown in the bottom margin are not always fully proven.

Thank you for the suggestion. We have now adjusted the Figure.

10. "MI" should be spelled out at its first appearance in the abstract (line 40).

It was adjusted in the text.

Possible typos: 1-Pearson (line p.97); Immunofluorescence-> Immunostaining assay (line 117).

It was corrected in the text.

Reviewer #4 (Remarks to the Author):

This manuscript by Gladka et al describes a potential novel role for Zeb2 in regulating myocardial function post-myocardial infarction via its control of the secretion of angiogenic molecules. Overall, the manuscript was quite well written and the data clearly presented. However, there are a few areas that need to be clarified.

We would like to thank the reviewer for the comments. We have attempted to address all issues raised by this reviewer to improve the quality of our work.

1. If these two new paracrine factors are essential for mediating Zeb2 effects, an antibody neutralization of the conditioned medium would demonstrate that they are critical for mediating Zeb2's pro-angiogenic effect. Also, what is the nature of Zeb2 regulation of these genes? Are there Zeb2 binding sites in their promoters for example?

We didn't use neutralizing antibodies, but alternatively, we made use of siRNA-mediated silencing of ZEB2, TMSB4 and PTMA in cardiomyocytes to demonstrate that the observed angiogenic phenotype is Zeb2-dependent and that this pro-angiogenic function of ZEB2 occurs via TMSB4 and PTMA regulation and secretion from cardiomyocytes. To do so, we performed a spheroid-based sprouting assay, which is a well-established method to study capillary-like tube formation of cultured endothelial cells. The advantage of this assay is the possibility to study angiogenesis in a 3D environment. We treated iPSC-derived cardiomyocytes with siRNA-control, siRNA-ZEB2, siRNA-TMSB4, siRNA-PTMA and combination of siRNA-TMSB4 and siRNA-PTMA (Figure 19a). We collected conditioned medium from treated cardiomyocytes and used it to treat HUVEC cells in sprouting assay (Figure 19a). We imaged multiple spheroids (Figure 19b) and quantified the total number of sprouts per spheroid as well as cumulative sprout length per spheroid. We observed that ZEB2 inhibition resulted in a significant decrease in the number of sprouts per spheroid (Figure 19c) and a non-significant decrease in cumulative sprout length per spheroid (Figure 19d). Inhibition of TMSB4 and PTMA separately did not affect the number and length of sprouts, but when inhibited together, they show similar results as ZEB2 inhibition.

We also check the effect of inhibition of ZEB2, TMSB4, PTMA4 and TMSB4 + PTMA together in cardiomyocytes (as described in Figure 20a) on endothelial cell proliferation. We treated HUVEC with conditioned medium and analyzed 48 hours later (Figure 20a). We analyzed EdU positive HUVEC cells and observed a significant decrease in EdU positive nuclei when treated with conditioned medium from ZEB2 inhibited cardiomyocytes (Figure 20b-c). Even though we did not reach the significance in the decrease of EdU positive HUVEC cells in TMSB4 and

PTMA treated conditions, we observed a very pronounced decrease in the expression of cell cycle genes in all conditions (**Figure 20d-f**). Additionally, we observed a decrease in *Pecam1*, *Angiogenin* and *Endoglin*, which are involved in angiogenesis and cell migration (**Figure 20g-i**). Together we showed that inhibition of ZEB2 in cardiomyocytes (and to lower extent individual factors TMSB4 and PTMA), influences endothelial cell sprouting, proliferation and contributes to inhibition of angiogenic response.

Regarding the regulation, indeed, based on oPOSSUM (which is a web-based system for the detection of over-represented conserved transcription factor binding sites), there are many predicted binding sites of ZEB2 in the mouse and human promoters of *Tmsb4* and *Ptma*. We screened proximal promoter of *Tmsb4* and *Ptma* 10 kb upstream and 10 kb downstream from the transcription starting site (TSS) with the following settings: Conservation cutoff: 0.40 and Matrix score threshold: 85%. As shown in Supplementary Figure 5J we identified multiple potential binding sites.

Additionally, we made use of existing Chip-seq dataset <https://www.encodeproject.org/targets/ZEB2-human/> and identified ZEB2 binding motifs in enhancer regions of both TMSB4 (**Figure 21a**) and PTMA (**Figure 21b**), indicating that ZEB2 protein directly binds to those enhancers causing an increase on TMSB4 and PTMA expression.

Figure 21
2. The scratch wound healing assay is a read out of both effects on cell migration and cellular proliferation. As such it should be complemented with more clear migration assays such as the Boyden chamber assays. As well, the effect on tube formation is a well-accepted *in vitro* readout of angiogenesis and its addition to the current study would strengthen the manuscript.

We performed a spheroid-based sprouting assay, which is a well-established method to study capillary-like tube formation of cultured endothelial cells. Some of the results were already described when answering question 1 of the reviewer (**Figure 19 and 20**). In short, we showed that treatment with conditioned medium from ZEB2-deleted cardiomyocytes resulted in a significant decrease in the number of sprouts and a decrease in cumulative sprout length per spheroid (**Figure 19c-d**).

We also checked the effect of inhibition of ZEB2, TMSB4, PTMA4 and TMSB4 + PTMA together in cardiomyocytes (as described in **Figure 20**) on endothelial cell proliferation.

We also performed the same experiment where we looked at the sprout formation of endothelial cells when treated with conditioned medium from cardiomyocytes treated with AAV9-ZEB2, AAV9-TMSB4, AAV9-PTMA and combination of AAV9-TMSB4 and AAV9-PTMA (**Figure 22a**). Cells were additionally treated with VEGF. We imaged multiple spheroids (**Figure 22b**) and quantified a total number of sprouts per spheroid as well as cumulative sprout length per spheroid. We observed an increase of the total number of sprouts per spheroids and cumulative sprout length when treated with the conditioned medium from ZEB2 overexpressing cardiomyocytes in VEGF stimulated condition (**Figure 22c-d**). Additionally, we observed that the condition with TMSB4 alone could trigger the increase in the number and total length of sprouts (**Figure 22c-d**).

We also checked the effect of overexpression of ZEB2, TMSB4, PTMA and TMSB4 + PTMA together in cardiomyocytes on endothelial cell proliferation. We treated HUVEC with conditioned medium and analyzed 48 hours later (**Figure 23a**). We analyzed EdU positive HUVEC cells and observed a significant increase in EdU positive nuclei when treated with all conditions (**Figure 23b-c**).

Together we showed that inhibition and overexpression of ZEB2 in cardiomyocytes (and to the lower extent individual factors TMSB4 and PTMA) influences endothelial cell sprouting, proliferation and contribute to inhibition of angiogenic response.

Figure 22

Figure 23

3. Cardiac function is shown for the fl/fl mice versus control after MI but should be also shown prior to MI to show if there is any effect of strain/deletion on basal cardiac parameters.

The main goal of the study is to reveal the function of ZEB2 under ischemic conditions, therefore we excluded shams from all figures. We performed a baseline study to assess the effect of ZEB2 deletion and overexpression

under physiological conditions. Below we enclosed figures from the original manuscript. In **Supplementary Figure 2** we studied the baseline effect of cardiomyocyte-specific Zeb2 deletion and in **Supplementary Figure 6** we focused on cardiomyocyte-specific overexpression. Both studies indicated that there is no baseline morphological and functional effect in our gain- and loss-of-function models.

4. In addition to fractional shortening, ejection fraction is a critical clinical parameter of cardiac function and should be measured. The authors may also want to look at E/A ratio which is a measure of diastolic function.

We have now added EF measurements which are placed in the Supplementary Table 1, 2, 3, 4, 5 and 6 together with other echocardiographic measurements. Unfortunately, we did not measure E/A ratio in all studies.

5. Was cardiac fibrosis altered in this model? It should be discussed as it is known to affect cardiac function.

As suggested by the reviewer we did indeed observe difference in the fibrotic response in response to ZEB2 modulation. In line with observed impairment of infarct healing in *Zeb2* cKO mice, we also saw that the fibrotic scar in *Zeb2* cKO mice was less mature than in the wild type controls, as shown by the birefringence images (**Figure 24a-b**). In the *Zeb2* cKO hearts, the expression of *Periostin*, which is a marker of fibroblast activation, was significantly decreased (**Figure 24c**), indicating less fibroblast activation after *Zeb2* removal in cardiomyocytes. This was confirmed by the dramatic reduction in *Comp* expression (**Figure 24d**), which is a bone and a cartilage marker. We were unable to detect significant differences in the scar maturity in the *Zeb2* cTg hearts compared to control (**Figure 24e-f**). Both *Periostin* and *Comp* are known to be highly upregulated 14 days post-MI (**Figure 24g-h**). The effect of *Zeb2* on the fibrotic response needs more careful investigation and will be addressed in follow-up studies.

6. The cardiac specific deletion reduced total cardiac *Zeb2* by approximately 50% in the post-MI heart. What non-cardiomyocyte cells express *Zeb2* in this setting?

Using immunohistochemistry, we could also observe *ZEB2* expression in fibroblasts in the infarct region (**Figure 25a** – white arrows) and endothelial cells in the border zone and remote (**Figure 25b** – white stars).

7. Cell death should be measured in addition to looking at levels of Bcl-XL

In addition to Bcl-XL, we also looked at the expression of pro-survival markers (*Bcl2* and *Survivin*) in the *Zeb2* overexpressing as well as *Zeb2* KO mice post-injury. We observed that pro-survival markers were enhanced in *Zeb2* Tg mice, which we think is due to the observed increase in perfusion of the tissue and better survival of cardiomyocytes (Figure 26a-b). Those markers showed a downward trend in *Zeb2* cKO hearts after MI, indicating that *Zeb2* cKO mice are less protected against ischemia-induced apoptosis.

8. The phrase “trending decrease” should be removed from the results and the results more clearly describe no significant change. The possible trends could be brought up in the discussion.

We have adjusted it accordingly in the text.

9. Acronyms should be spelled out in the abstract.

Thank you, we have now adjusted it in the abstract.

REVIEWER COMMENTS

Reviewer #2 (Remarks to the Author):

Gladka and colleagues have responded to my previous concerns. While their response is quite detailed and contains many Figures (1-9), the authors unfortunately do not indicate exactly where and how they have changed or amended their manuscript accordingly. This is usually done by providing in the response letter the changes that have been made in verbatim (plus the corresponding page and line numbers in the revised Ms.). This reviewer is unable to print and place side by side the previous Ms., the response, and the new Ms. (which is provided as 30 distinct downloadable items). The authors need to provide me, please, with that information for all my previous questions.

This general comment aside, I am still not happy by the way the angiogenic phenotype is described. The authors now indicate that angiogenesis was assessed in the remote myocardium. They observed a reduced capillary density in the remote myocardium of Zeb conditional KO compared to WT mice (Figure 2n). They need to assess capillary density also under sham-operated conditions; Zeb KO mice may have a reduced capillary density in the myocardium already at baseline, i.e. after sham surgery (as suggested by Figure 1 in the response). This info is missing from Figure 2 and Suppl. Table 1 (and from Figure 3 in the response). Figure 2 in the response (while presenting only data on Pecam mRNA expression in WT mice - I had hoped for a comparison with KO) confirms, what is long known, that angiogenesis after MI occurs in the border zone, not in the remote myocardium. This also suggests to me that reduced capillary density in Zeb cKO remote myocardium (Figure 2n) may correspond to a similar decrease already under sham conditions.

Similarly, the authors need to report if Zeb TG mice have increased capillary density in the myocardium already under sham conditions, which is what I suspect (not shown in Figure 7 and Suppl. Table 3). Both in the KO and TG models, Zeb deletion and overexpression occurs long before the MI (as driven by the aMHC promoter).

In many places of the Ms. the authors refer to 'infarct/wound healing' or 'cardiac repair' (e.g. lines 153-158; 343-346, 351, 384-385). They need to be clear in the Ms. that Zeb effects (KO and TG) were studied and observed in the remote myocardium. I don't understand how effects in the remote myocardium may affect wound healing/repair/scar formation (e.g. thinner scars in KO and thicker scars in TG – Figures 2f and 7g), unless you assume that myocardial tissue morphology was already different (e.g. different capillarization) before the MI (see my last 2 points).

Figure 5g,h (in the response) depicts striking differences in post MI mortality in KO and TG mice; the authors attribute this to the surgeries having been performed by 'different animal surgeons'. Mouse strain differences may be an alternative explanation. I could not find that information in the Supplement.

Minor: response to Additional points e): how can there ever be a 'definitive trend' with an n=3? What is a 'definitive trend'?

Reviewer #3 (Remarks to the Author):

I appreciate the substantial efforts of the authors to revise the manuscript. With the substantial and detailed extensive analyses and enriched discussion, the comprehensive view of this mechanism has become much clearer. Now, I found all the concerns which I raised in the previous round of the review have been satisfactorily addressed. I sincerely hope this work should lay the groundwork for future studies further deepening the current understandings how the damage response of cardiomyocytes are controlled, and eventually leading to the development of better therapeutic means.

Reviewer #4 (Remarks to the Author):

The authors have done a thorough job in responding to previous concerns.

Nature Communication NCOMMS-19-30116

"Cardiomyocytes stimulate angiogenesis after ischemic injury in a ZEB2-dependent manner"

We are grateful to the editors and reviewers for their comments and we tried to address all remaining issues of Reviewer #2 point-by-point.

Reviewer #2 (Remarks to the Author):

Gladka and colleagues have responded to my previous concerns. While their response is quite detailed and contains many Figures (1-9), the authors unfortunately do not indicate exactly where and how they have changed or amended their manuscript accordingly. This is usually done by providing in the response letter the changes that have been made in verbatim (plus the corresponding page and line numbers in the revised Ms.). This reviewer is unable to print and place side by side the previous Ms., the response, and the new Ms. (which is provided as 30 distinct downloadable items). The authors need to provide me, please, with that information for all my previous questions.

We apologize for not including the response letter containing addressed changes. We will provide such a response letter containing all changes made in the manuscript in response to the reviewer's comments (first and second revision).

Reviewer #2 revision #1 question 3

3) Infarct scar formation in ZEB2-deleted or -overexpressing mice needs to be better described (e.g. scar size in [%] of LV circumference, scar thickness, rupture rates, impact on mortality).

We now included the data (revision#1 Figure 5a and 5c) in **Supplementary Figure 3b-c** and (revision#1 Figure 5b and 5d) in **Supplementary Figure 9c-d**.

We also adjusted it in the text:

Lines 173-175: We also observed an increase in infarct size in *Zeb2* cKO compared with *Zeb2* fl/fl mice, assessed by the percentage of the infarcted area of the total LV circumference (**Supplementary Fig. 3b-c**).

Lines 432-435: Echocardiography showed an improvement in cardiac function (FS), a decline in left ventricular internal diameter (LVIDs) and smaller infarct size in mice overexpressing ZEB2 in cardiomyocytes (**Fig. 7e-f, Supplementary Fig. 9b-d and Table 4**).

Reviewer #2 revision #1 question 5

5) The authors show that conditioned supernatants from ZEB2 expressing cardiomyocytes promote angiogenesis. Do they have neutralizing antibodies at hand (or can they use another experimental setup) to demonstrate that this is a TMSB4 and/or PTMA-dependent effect?

We now included the data (revision#1 Figure 7a-d and Figure 8a-f) in **Figure 5a-i**.

We also adjusted the text:

Lines 351-392: Additionally, we performed sprout formation and proliferation assays to further validate the importance of ZEB2 in different aspects of the angiogenic response. To this end, we treated iPSC-derived cardiomyocytes with siRNA-control, siRNA-ZEB2, siRNA-TMSB4, siRNA-PTMA and a combination of siRNA-TMSB4 and siRNA-PTMA (**Fig. 5a**). Next, we collected conditioned medium from treated cardiomyocytes and used it to treat HUVECs (**Fig. 5a**). In doing so observed that ZEB2 inhibition in cardiomyocytes resulted in a significant decrease in the number of sprouts per spheroid and a non-significant decrease in cumulative sprout length per spheroid (**Fig. 5 b-d**). Inhibition of TMSB4 and PTMA separately did not affect the number and length of sprouts, but when

inhibited together, it showed a comparable effect as ZEB2 inhibition (Fig. 5 b-d). We also analyzed endothelial cell proliferation and observed a significant decrease in EdU positive nuclei when treated with conditioned medium from ZEB2-inhibited cardiomyocytes (Fig. 5 e-f). Even though we did not reach significance in the decrease of EdU positive HUVECs in TMSB4 and PTMA-treated conditions, we did observe a very pronounced decrease in the expression of cell cycle genes in all conditions (Fig. 5 g-i).

Reviewer #2 revision #1 additional comment c)

c) The discussion is rather superficial and largely recapitulates the results. The authors may want to provide more info on TMSB4 and PTMA. Are these genes related? Sequence similarities? What is known from previous studies regarding their tissue / cellular expression patterns and role(s) in tissue/heart angiogenesis under normal conditions or after stress?

We now added additional information about TMSB4 and PTMA in the discussion:

Lines 632-639: TMSB4 treatment is already being used to promote wound repair in skin, cornea and heart in a series of ongoing clinical trials.³² In the heart, studies have shown that TMSB4 stimulates the formation of new cardiomyocytes, which originate from the epicardial layer³⁰, and promotes neovascularization.³¹ Additionally, TMSB4 has been shown to play anti-inflammatory, antioxidant and antifibrotic effects during liver injury in mice.³⁴ On the other hand, PTMA has been shown to stimulate cardiac endothelial cell migration, angiogenesis and wound healing²⁰. It has also been recently proven to protect from retinal ischemic damage.³⁵

Reviewer #2 revision #1 additional comment f)

f) Line 206: a 'trend' usually implies a P value <0.10. These passages need to be reworded.

We reworded the sentence, lines 279-282: While the expression of many factors did not change in *Zeb2* cKO mice post-MI, we observed a significant downregulation for *Tmsb4*, *Ptma*, *Rack1* and *Dynll1* (Fig. 4a-b, Supplementary Fig. 6a-i).

Reviewer #2 revision #1 additional comment g)

g) Line 274: the AAV9 vector used to overexpress ZEB2 utilizes a CMV promotor and does not deliver ZEB2 specifically to cardiomyocytes (Suppl. Fig. 8). This should be noted as a limitation.

We included it now in the discussion, lines 648-651: Targeted delivery using a cardiomyocyte-specific promoter would further improve the specificity of the therapy and allow for a more directed treatment in order to prevent undesired effects in other organs.

This general comment aside, I am still not happy by the way the angiogenic phenotype is described. The authors now indicate that angiogenesis was assessed in the remote myocardium. They observed a reduced capillary density in the remote myocardium of *Zeb* conditional KO compared to WT mice (Figure 2n). They need to assess capillary density also under sham-operated conditions; *Zeb* KO mice may have a reduced capillary density in the myocardium already at baseline, i.e. after sham surgery (as suggested by Figure 1 in the response). This info is missing from Figure 2 and Suppl. Table 1 (and from Figure 3 in the response).

We want to thank the reviewer for these additional questions. We quantified PECAM1 positive areas in hearts under baseline conditions and indeed, there is a decrease in capillary density in *Zeb2* cKO mice compared to *Zeb2* fl/fl sham mice (Figure 1a-b). Additionally, the expression of several angiogenic markers are also decreased in *Zeb2*fl/fl vs *Zeb2* cKO hearts (although not significantly n=4).

We now also included these data in **Supplementary Figure 3d-j** and addressed this in the text: Lines 194-196: A decrease in capillary density and expression of endothelial markers was also observed in Zeb2 cKO mice after sham, implying a vascular difference at baseline (**Supplementary Fig. 3d-j**).

Figure 1

Figure 2 in the response (while presenting only data on Pecam mRNA expression in WT mice - I had hoped for a comparison with KO) confirms, what is long known, that angiogenesis after MI occurs in the border zone, not in the remote myocardium. This also suggests to me that reduced capillary density in Zeb cKO remote myocardium (Figure 2n) may correspond to a similar decrease already under sham conditions.

We apologize we did not interpret the question correctly. We studied the effect of Zeb2 deletion only at a one-time point (14days post-MI). However, based on the observations that the capillary density and endothelial gene expression seems to be already lower in mutant mice under sham conditions, we speculate that this is also true for all intermediate timepoints.

Similarly, the authors need to report if Zeb TG mice have increased capillary density in the myocardium already under sham conditions, which is what I suspect (not shown in Figure7 and Suppl. Table 3). Both in the KO and TG models, Zeb deletion and overexpression occurs long before the MI (as driven by the aMHC promoter).

We now performed additional quantification of capillary density in the Zeb2 cTg vs. Zeb2 WT hearts under sham conditions. We see a small increase in PECAM1 positive areas in the Zeb2 cTg hearts (Figure 2a-b). We also looked at the expression of angiogenic markers in sham Zeb2 WT vs. Zeb2 cTg hearts and observed that many of them are increased, indicating that there are more endothelial cells in Zeb2 cTg mice already at baseline.

We now also included these data in **Supplementary Figure 9e-k** and addressed this in the text: Lines 442-445: Additionally, we already observed a mild increase in capillary density and endothelial markers under sham conditions, which can potentially contribute to the enhanced angiogenesis post-MI (**Supplementary Fig. 9e-k**).

We also comment on this in the Discussion, lines 645-648: Although genetic deletion and overexpression of ZEB2 in cardiomyocytes have minor effects on capillary density under baseline conditions, therapeutic delivery of ZEB2 post-injury is sufficient to restore cardiac function by stimulating angiogenesis, which in turn improves cardiac repair.

Figure 2

In many places of the Ms. the authors refer to ‘infarct/wound healing’ or ‘cardiac repair’ (e.g. lines 153-158; 343-346, 351, 384-385). They need to be clear in the Ms. that Zeb effects (KO and TG) were studied and observed in the remote myocardium. I don’t understand how effects in the remote myocardium may affect wound healing/repair/scar formation (e.g. thinner scars in KO and thicker scars in TG – Figures 2f and 7g), unless you assume that myocardial tissue morphology was already different (e.g. different capillarization) before the MI (see my last 2 points).

We want to thank the reviewer for pointing this out. Indeed, we should not omit the fact that the border zone plays a primary role in cardiac repair. We saw a smaller effect in the border zone in the expression of angiogenic markers because, during tissue collection, we do not isolate the border zone alone but combine it with the infarcted area. This might dilute the signals coming specifically from the border zone. Now we also quantified capillary density in the border zone and observed that indeed there is a significant decrease in PECAM1 positive area in the Zeb2 cKO hearts when compared to the Zeb2 fl/fl post-MI (**Figure 3a-b**). Conversely, we observe an increase in angiogenic signals in the border zone from Zeb2 cTg hearts when compared to the border zone of Zeb2 WT hearts (**Figure 3c-d**).

We now also included these data in **Figure 2n, 2b** and **Figure 7m, 7o** and addressed this in the text, lines 190-194: We observed a significant reduction in the endothelial markers *Pecam1* and *Vegf1* in the infarcted *Zeb2* cKO hearts when compared to *Zeb2* fl/fl controls (**Fig. 2k-l**), which corresponded with an overall decrease in the percentage of PECAM1 positive vessels in the remote and border zone (**Fig. 2m-p**).

Lines 439-442: This corresponded with an increase in *Pecam1* at both the mRNA and protein levels (**Fig. 7l-n**) and an overall increase in the PECAM1 positive area in the *Zeb2* cTg hearts in the remote and border zone (**Fig. 7o-p**).

Figure 3

Figure 5g,h (in the response) depicts striking differences in post MI mortality in KO and TG mice; the authors attribute this to the surgeries having been performed by 'different animal surgeons'. Mouse strain differences may be an alternative explanation. I could not find that information in the Supplement.

The strain of mice used in this manuscript is the same for all studies: C57Bl6J, obtained from Charles River Laboratories.

Minor: response to Additional points e): how can there ever be a 'definitive trend' with an n=3? What is a 'definitive trend'?

“Definitive trend” has been removed from the manuscript.

REVIEWERS' COMMENTS

Reviewer #2 (Remarks to the Author):

Thank you for addressing my remaining concerns.